# Universal Video Temporal Grounding with Generative Multi-modal Large Language Models

**Zeqian Li**[1], **Shangzhe Di**[1,2], **Zhonghua Zhai**[2],

**Weilin Huang**[2], **Yanfeng Wang**[1], **Weidi Xie**[1]

[1]SAI, Shanghai Jiao Tong University    [2]ByteDance Seed

https://lzq5.github.io/UniTime

## Abstract

This paper presents a computational model for universal video temporal grounding, which accurately localizes temporal moments in videos based on natural language queries (*e.g.,* questions or descriptions). Unlike existing methods that are often limited to specific video domains or durations, we propose **UniTime**, a robust and universal video grounding model leveraging the strong vision-language understanding capabilities of generative Multi-modal Large Language Models (MLLMs). Our model effectively handles videos of diverse views, genres, and lengths while comprehending complex language queries. The key contributions include: (i) We consider steering strong MLLMs for temporal grounding in videos. To enable precise timestamp outputs, we incorporate temporal information by interleaving timestamp tokens with video tokens. (ii) By training the model to handle videos with different input granularities through adaptive frame scaling, our approach achieves robust temporal grounding for both short and long videos. (iii) Comprehensive experiments show that UniTime outperforms state-of-the-art approaches in both zero-shot and dataset-specific finetuned settings across five public temporal grounding benchmarks. (iv) When employed as a preliminary moment retriever for long-form video question-answering (VideoQA), UniTime significantly improves VideoQA accuracy, highlighting its value for complex video understanding tasks.

## 1 Introduction

Understanding long videos presents fundamental challenges due to complex temporal dependencies and varying information granularity. Recent approaches [30, 42] typically adopt a two-stage framework: first localizing relevant temporal segments, then reasoning over the retrieved content. This retrieval-then-reasoning paradigm offers a structured and interpretable framework for tackling intricate video understanding tasks.

However, existing grounding models perform unsatisfactorily, often struggling to generalize across diverse video domains [50]. In this context, we advocate for developing a universal temporal grounding model, that is capable of accommodating videos across a wide range of viewpoints (egocentric and exocentric perspectives), topics (from cooking and daily activities to sports, movies, and games), and durations (from brief clips to multi-hour videos). Such universality is crucial for real-world deployment, where video data is inherently heterogeneous and user queries are highly diverse.

Research on temporal grounding falls into two paradigms. The first includes discriminative or dual-encoder-based approaches [4, 24], which typically rely on dataset-specific fine-tuning and lightweight architectures, such as DETR-like frameworks. While computationally efficient, these models often struggle to generalize to unseen content due to limited language encoding capabilities, hindering the understanding of complex or user-driven queries, and constrained visual representations. In

39th Conference on Neural Information Processing Systems (NeurIPS 2025).

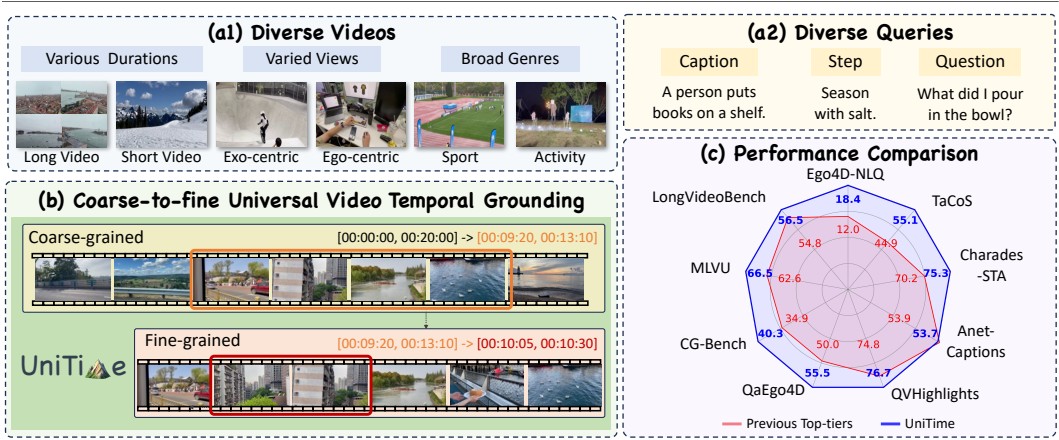

Figure 1: The **UniTime** framework empowers MLLMs with advanced universal temporal grounding capabilities. (a) UniTime can handle diverse videos with various views, genres, and durations, as well as comprehend complex language queries. (b) UniTime achieves universal temporal grounding through a coarse-to-fine approach. (c) Performance comparison on temporal grounding and video question answering benchmarks demonstrates the superior capabilities of UniTime.

contrast, recent work leverages Multi-modal Large Language Models (MLLMs) [34, 62], which excel at language understanding and demonstrate promising results on video temporal grounding. However, their applicability to long videos is hindered by short context windows and high memory demands, making them impractical for processing video inputs spanning tens of thousands of tokens.

In this paper, we further explore the potential of MLLMs towards universal temporal grounding in videos. Our proposed model, termed as **UniTime**, encodes temporal information by interleaving timestamp tokens with video tokens, enabling precise boundary generation for natural language queries. To handle videos of varying lengths, we introduce an adaptive frame scaling mechanism that adjusts temporal and spatial granularity based on video duration. Specifically, for long videos, the model can be adopted for multi-stage inference-time computation: initial coarse-grained sampling enables segment-level grounding, followed by fine-grained sampling within candidate regions to refine boundaries. This iterative process progressively improves grounding precision. To improve training efficiency, we adopt a video-centric training paradigm that concatenates multiple queries and their corresponding segments per video, allowing the model to process them in a single pass. This minimizes the computational overhead of autoregressive decoding over repeated video tokens.

To rigorously evaluate efficacy, we pre-train **UniTime** on diverse, high-quality temporally grounded video datasets and assess its performance on two key tasks: **temporal grounding** (in both zero-shot and fine-tuning settings) and downstream **video question answering** (VideoQA) for long videos. For temporal grounding, we conduct experiments on two long-video and three short-video benchmarks, where our model consistently outperforms existing methods in both straightforward and complex retrieval scenarios. For downstream VideoQA, we evaluate on four widely used long-video benchmarks using a retrieval-augmented protocol: first, our framework localizes the relevant video segment; then, an off-the-shelf VideoQA model answers the query based on the localized segment. Compared to prior approaches, our method demonstrates superior generalization, delivering more accurate and comprehensive video understanding, especially for long videos.

The remainder of the paper is organized as follows: Section 2 formulates the temporal grounding task and details our proposed method. Section 3 presents ablation studies and comparisons to validate our approach. Section 4 provides an overview of the relevant literature. Overall, our method achieves improved performance on both temporal grounding and long-video VideoQA benchmarks.

## 2 Methods

In this section, we introduce **UniTime**, a novel framework for universal video temporal grounding, adapted from pre-trained Multi-modal Large Language Models (MLLMs). Our approach achieves precise temporal grounding across videos of diverse scenes, genres, and durations, and employs an iterative coarse-to-fine strategy to handle long videos. In the following, Section 2.1 formalizes the

problem setting. Section 2.2 presents our architecture and its core modules, which support flexible temporal grounding at arbitrary granularities via inference-time computation. Section 2.3 details an efficient training paradigm that equips MLLMs with universal temporal grounding capabilities.

## 2.1 Problem Formulation

Given an untrimmed video sequence $\mathcal{V} = \{f_1, \ldots, f_{N_f}\} \in \mathbb{R}^{N_f \times H \times W \times 3}$ sampled at timestamps $\mathcal{T} = \{t_1, \ldots, t_{N_f}\}$ and a free-form text query $\mathcal{Q}$ (either a complex question or a plain description), our task is to identify all temporal moments $\mathcal{Y} = \{(s_1, e_1), \ldots, (s_K, e_K)\}$ that semantically match the query, where each $s_k, e_k \in \mathcal{T}$ denotes a start and end timestamp, respectively:

$$\mathcal{Y} = \Phi_{\text{UniTime}}(\mathcal{V}, \mathcal{T}, \mathcal{Q}).$$

With this formulation, temporal grounding is achieved by identifying the frames most relevant to the query and retrieving their timestamps. By adjusting the granularity of $\mathcal{T}$, our design supports multi-scale temporal predictions, enabling efficient coarse-to-fine grounding for long videos through inference-time computation, as shown in Figure 1(b).

## 2.2 Universal Temporal Grounding Model

The core innovation of UniTime lies in its capability to perform predictions at multiple temporal granularities for videos of diverse lengths. The model consists of three key components: (i) adaptive frame scaling for constructing multi-granular video inputs, (ii) timestamp-interleaved sequence construction enabling multi-scale temporal window prediction, and (iii) multi-scale prediction for coarse-to-fine temporal grounding during inference, as illustrated in Figure 2.

**Adaptive Frame Scaling.** A central challenge in universal temporal grounding is handling videos with highly variable durations, ranging from a few seconds to several hours. Applying fixed temporal and spatial resolutions across such diverse inputs is inefficient and often impractical, especially for MLLMs constrained by limited context windows and GPU memory.

To address this, we propose an adaptive frame scaling mechanism that dynamically adjusts per-frame token budget ($N_{\text{res}}$) based on video duration, allocating high spatial resolution to short videos and low resolution to long videos. Specifically, given a video $\mathcal{V}$ with constant frame rate ($N_f$ frames in total), each frame is allocated $N_{\text{res}} = \lfloor N_{\text{total}}/N_f \rfloor$ tokens, where $N_{\text{total}}$ is the total token budget. To support varying spatial granularities, we set two thresholds, $N_f^{\text{short}}$ and $N_f^{\text{long}}$, as the maximum number of frames for short and long videos, respectively. Frames are then processed as:

$$\mathbf{V}_i = \begin{cases} \phi_{\text{project}}\big(\phi_{\text{vision}}(\psi_{\text{resize}}(f_i))\big) \in \mathbb{R}^{N_{\text{res}} \times d}, & N_f < N_f^{\text{short}}. \\ \psi_{\text{compress}}\big(\phi_{\text{projector}}\big(\phi_{\text{vision}}(f_i)\big)\big) \in \mathbb{R}^{N_{\text{res}} \times d}, & N_f^{\text{short}} \leq N_f < N_f^{\text{long}}. \end{cases}$$

where $\phi_{\text{vision}}(\cdot)$ is the vision encoder, which supports dynamic input resolutions with a fixed patch size of $P$, and $\phi_{\text{project}}(\cdot)$ is the projection module. $\psi_{\text{resize}}(\cdot)$ denotes the frame resizing function, which adjusts the spatial resolution of each frame so that it can be partitioned into $N_{\text{res}}$ patches of size $P \times P$ pixels. $\psi_{\text{compress}}(\cdot)$ denotes token compression via bilinear interpolation, yielding $N_{\text{res}}$ tokens as the final representation. Unlike frame resizing, which may degrade spatial details, token compression reduces redundancy at the token level while better preserving key semantic information, especially for long videos with low resolutions per frame. When the number of frames exceeds $N_f^{\text{long}}$, the video is divided into multiple clips of length $N_f^{\text{long}}$ for divide-and-conquer processing. The ablation experiments of token budget, thresholds and the input processing strategy can be found in Appendix E.3 and E.1.

**Discussion.** Existing MLLMs typically employ fixed frame sizes and token allocations [33, 68, 25], which limits their ability to encode long videos effectively. To handle such inputs, these models must resort to sparse temporal sampling, resulting in significant loss of detailed visual content. In contrast, our adaptive strategy dynamically employs higher spatial resolutions for short videos and lower ones for longer videos. This allows for efficient and scalable modeling across diverse video lengths, maximizing the utility of the available token budget and model capability.

**Timestamp-Interleaved Sequence Construction.** To associate frames with their timestamps, we encode temporal information by interleaving timestamp tokens between visual tokens. For a frame $f_i$

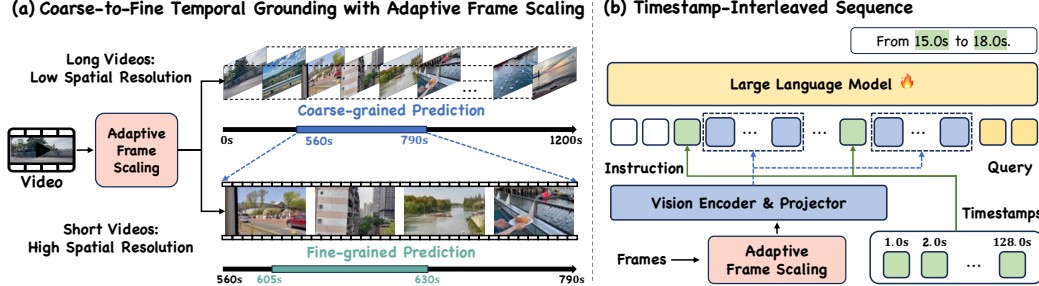

Figure 2: Overview of the proposed **UniTime** framework. (a) UniTime achieves universal temporal grounding by leveraging adaptive frame scaling to construct multi-scale video inputs and then generate multi-scale predictions, allowing robust grounding across diverse video durations. (b) Within the model architecture, UniTime constructs an interleaved sequence of timestamps and scaled frame features, which, combined with the language query, is fed into the LLM and then identifies the corresponding temporal interval from the timestamp tokens.

sampled at timestamp $t_i$, the timestamp is represented as free text $\tau_i =$ "`timestamp:` $t_i$ `seconds`", and inserted before visual tokens of $f_i$, forming an interleaved sequence:

$$\mathcal{S} = [\mathbf{T}_1; \mathbf{V}_1; \mathbf{T}_2; \mathbf{V}_2; \ldots; \mathbf{T}_{N_f}; \mathbf{V}_{N_f}; \mathcal{Q}], \quad \mathbf{T}_i = \phi_{\text{tokenizer}}(\tau_i) \quad (1)$$

where $\phi_{\text{tokenizer}}(\cdot)$ is the tokenizer. This interleaved sequence is input to the large language models (LLMs), which generates temporal boundaries for a given query $Q$, with the output formatted as $\mathcal{A} =$ "`From` $s_k$ `seconds to` $e_k$ `seconds`". Note that, unlike conventional methods that predict the exact moment in the original video, our model instead predicts the minimal timestamp interval $(s_k, e_k)$ from the sampled timestamps set $\mathcal{T}$. That is to say, we leverage the retrieval capabilities of MLLMs to read out the inserted timestamp tokens, rather than decoding dense position encodings. For a comparison of these temporal information encoding approaches, refer to the Appendix E.2.

It is worth noting that direct fine-grained temporal interval prediction on coarse-grained long video inputs is inherently unreliable, as the lack of detailed spatial information often results in ambiguous or inaccurate predictions. This highlights the necessity of adopting predictions at different scales for videos with varying granularities. Our framework addresses this challenge by offering inherent flexibility in timestamp token insertion, making it adaptable to varying granularities. Specifically, timestamp tokens can be interleaved at different scales: before each frame for fine-grained temporal predictions, or before fixed-length segments for coarse-grained temporal reasoning.

We divide the input video into $N_s$ segments $\{\mathbf{S}_j\}_{j=1}^{N_s}$, each comprising $\mathbf{L}_s$ frames. Here, for the $j^{th}$ segment, $\mathbf{S}_j = [\mathbf{V}_{s_j}; \mathbf{V}_{s_j+1}; \cdots; \mathbf{V}_{s_j+\mathbf{L}_s-1}] \in \mathbb{R}^{\mathbf{L}_s \times N_{\text{res}} \times d}$, where $s_j$ is the start index of the segment and we use the corresponding timestamp $t_{s_j}$ at the beginning of each segment to represent its temporal location. Note that we prepend only a single timestamp text token to each segment, thereby providing coarse-grained temporal supervision. We also feed the interleaved timestamps, visual tokens, and the query into the LLM to retrieve the target segment.

$$\mathcal{S} = [\mathbf{T}_1; \mathbf{S}_1; \mathbf{T}_2; \mathbf{S}_2; \cdots; \mathbf{T}_{N_s}; \mathbf{S}_{N_s}; \mathcal{Q}], \quad \mathbf{T}_i = \phi_{\text{tokenizer}}(\tau_{s_j}) \quad (2)$$

**Discussion.** Here, explicitly providing timestamps offers several crucial advantages: (i) **Improved temporal grounding**. These explicit timestamp tokens help the LLM effectively associate visual content with its corresponding temporal position in the video, allowing it to generalize to temporal grounding tasks. (ii) **Alignment-free integration.** As the inserted timestamps are textual, they naturally reside in the language space and do not require realignment. In contrast, methods based on learned vectors necessitate additional embedding alignment, thereby complicating the LLM's understanding of temporal information. (iii) **Generalisable temporal reasoning.** When combined with adaptive frame scaling, explicitly provided timestamp tokens allow the model to directly refer to the temporal cues, potentially enabling extrapolation. For example, as demonstrated in Section 3.7, this design allows the model to be trained on short or medium-length videos (*e.g.*, around 6 minutes), while still generalizing effectively to much longer videos (*e.g.*, 20 minutes) during inference on downstream tasks. (iv) **Model-agnostic design.** This method is plug-and-play and can be easily incorporated into diverse model architectures with minimal modifications, which is validated in

Table 1: **Statistics of temporal grounding datasets.** The datasets in Part I are used exclusively for pre-training, while those in Part II are the training sets of benchmarks.

| | Dataset | #Videos | #Queries | Video Len. | Moment Len. | Views | Domain |
|---|---|---|---|---|---|---|---|
| | NaQ [46] | 4.9K | 1031K | 413s | 1.1s | Ego | Open |
| | DiDeMo [1] | 8.4K | 33.0K | 29s | 7.5s | Exo | Open |
| | QuerYD [38] | 819 | 5.7K | 278s | 13.6s | Ego & Exo | Open |
| I | HiRest [61] | 526 | 4.0K | 263s | 18.9s | Ego & Exo | Open |
| | COIN [53] | 11.8K | 46.4K | 145s | 14.9s | Ego & Exo | Open |
| | Momentor [45] | 6.7K | 136.4K | 403s | 49.5s | Ego & Exo | Open |
| | YouCook2 [72] | 1.2K | 9.6K | 316s | 19.7s | Ego & Exo | Cooking |
| | Ego4D-NLQ [11] | 1.0K | 10.0K | 495s | 8.2s | Ego | Open |
| | TACoS [47] | 75 | 9.8K | 287s | 27.9s | Ego & Exo | Cooking |
| II | Charades-STA [51] | 4.8K | 11.2K | 31s | 8.1s | Exo | Activity |
| | QVHighlights [24] | 7.1K | 7.2K | 150s | 24.6s | Ego & Exo | Vlog/News |
| | ANet-Captions [22] | 10K | 37.4K | 118s | 36.0s | Ego & Exo | Activity |

Section 3.4, eliminating the need for architecture-specific adjustments such as custom positional encoding schemes [12, 2].

**Universal Temporal Grounding with Multi-stage Inference.** Building on these designs, our framework achieves universal temporal grounding by processing input frames at different spatial resolutions and generating temporal predictions at appropriate granularities, according to video duration. For long videos, we employ a hierarchical, multi-stage inference strategy. Specifically, if a video exceeds $N_f^{\text{long}}$ frames, it is partitioned into multiple clips, each of length $N_f^{\text{long}}$. Segment retrieval is first performed within each clip to identify potentially relevant segments. The retrieved segments from all clips are then aggregated and subjected to further segment retrieval, and this process can be repeated recursively if necessary. Finally, fine-grained grounding is conducted within the final selected segments. As evidenced by Section 3.5, this flexible and hierarchical inference process ensures accurate and efficient temporal grounding across diverse video durations.

## 2.3 Training

In this section, we first describe the training objective for temporal grounding, followed by a detailed explanation of the hierarchical data construction for multi-granularity training. We then present a video-centric training mechanism to improve training efficiency.

**Training Loss.** The model is trained with an auto-regressive objective over mini-batches. For each training sample $(\mathcal{V}, \mathcal{T}, \mathcal{Q}, \mathcal{Y})$, *i.e.,* the video, timestamps, text query, and target, the prompt sequence $\mathcal{S}$ is constructed based on Equation 1 or 2 for short and long videos, respectively. Then, the model minimizes the negative log-likelihood over the target tokens only:

$$\mathcal{L}(\mathcal{S}, \mathcal{Y}) = -\sum_{i=1}^{N_y} \log P(y_t \mid \mathcal{S}, y_{<i}; \theta)$$

where $N_y$ is the target length.

**Training Data Construction.** To support temporal grounding at multiple granularities, we construct training data from videos of varying durations, using full videos for coarse-grained supervision and randomly sampling shorter segments containing ground-truth moments for fine-grained training. All samples are combined for joint training. Since this process naturally yields more short videos than long ones, we balance the data distribution by replicating long video samples with a factor $N_{\text{rep}} > 1$.

**Video-centric Training.** Given a dataset comprising paired videos, text queries, and corresponding moments $(\mathcal{V}^{(k)}, \mathcal{Q}^{(k)}, \mathcal{M}^{(k)})$, previous methods typically employ *query-centric* sampling: first sampling a query, then loading its associated video. This approach is highly inefficient when multiple queries correspond to the same video, as it introduces (i) an I/O bottleneck from repeatedly loading long videos and (ii) redundant computation due to re-encoding identical video content.

To address this, we employ *video-centric* sampling, wherein each video $\mathcal{V}^{(k)}$ is sampled first, and all associated query-answer pairs are grouped into a single input sequence. This strategy minimizes

Table 2: Statistics of benchmarks. The benchmarks encompass three representative tasks – Video Temporal Grounding, Grounded VideoQA, and General VideoQA. The reported statistics are based on the actual numbers of videos and queries used in our experiments.

| Benchmark | #Videos | #Queries | Video Len. | Moment Len. | Views | Domain |
|---|---|---|---|---|---|---|
| Ego4D-NLQ [11] | 415 | 4.6K | 500s | 10.7s | Ego | Open |
| TACoS [47] | 25 | 4.0K | 368s | 31.9s | Ego & Exo | Cooking |
| Charades-STA [51] | 1.3K | 3.7K | 29s | 7.8s | Exo | Activity |
| QVHighlights [24] | 1.5K | 1.5K | 150s | 32.2s | Ego & Exo | Vlog/News |
| ANet-Captions [22] | 4.9K | 17.0K | 118s | 40.2s | Ego & Exo | Activity |
| QaEgo4D [3] | 148 | 500 | 487s | 13s | Ego | Open |
| CG-Bench [5] | 1.2K | 12K | 1661.1s | 19.8s | Ego & Exo | Open |
| MLVU [71] | 1.1K | 2.2K | 755.7s | - | Ego & Exo | Open |
| LongVideoBench [57] | 618 | 1.2K | 574.9s | - | Ego & Exo | Open |

Table 3: **Comparison with SoTA methods on video temporal grounding benchmarks.** SP denotes the dataset-specific fine-tuning setting, and Full refers to the universal pre-training setting. Best results are in **bold**. Improvements of UniTime-Full over SoTA are highlighted in (green).

(a) Long-video temporal grounding benchmarks.

| Method | Ego4D-NLQ | | | TaCoS | | |
|---|---|---|---|---|---|---|
| | R1@.3 | R1@.5 | mIoU | R1@.3 | R1@.5 | mIoU |
| 2D-TAN [66] | 5.04 | 2.02 | 3.53 | 45.61 | 35.77 | - |
| VSLNet [65] | 10.84 | 6.81 | 8.83 | 35.54 | 23.54 | 24.99 |
| RGNet [13] | 18.28 | 12.04 | 12.41 | - | - | - |
| SnAG [37] | 15.87 | 11.26 | - | 56.44 | 44.86 | - |
| SG-DETR [10] | - | - | - | 56.71 | 44.70 | 40.90 |
| LD-DETR [70] | - | - | - | 57.61 | 44.31 | 40.30 |
| UniVTG [27] | 7.28 | 3.95 | 4.91 | 51.44 | 34.97 | 33.60 |
| UniVTG w/PT | 11.74 | 7.54 | 7.88 | 56.11 | 43.44 | 38.63 |
| Qwen2-VL-7B [56] | 0.48 | 0.17 | 0.46 | 4.14 | 1.27 | 2.64 |
| Qwen2.5-VL-7B [2] | 1.11 | 0.48 | 0.82 | 7.66 | 3.35 | 5.29 |
| UniTime-SP | 24.79 | 16.83 | 17.25 | 61.18 | 48.31 | 45.02 |
| UniTime-Full | **27.09** | **18.41** | **18.80** | **66.91** | **55.14** | **50.00** |
| | (+8.81) | (+6.37) | (6.39) | (+9.30) | (+10.44) | (+9.10) |

(b) Short-video temporal grounding benchmarks.

| Method | Charades-STA | | ANet-Captions | | QVHighlights | |
|---|---|---|---|---|---|---|
| | R1@.5 | R1@.7 | R1@.5 | R1@.7 | R1@.5 | R1@.7 |
| UnLoc-L [58] | 60.80 | 38.40 | 48.30 | 30.20 | 66.10 | 46.70 |
| SG-DETR [10] | 70.20 | 49.50 | - | - | 72.20 | 56.60 |
| LD-DETR [70] | 62.58 | 41.56 | - | - | 66.80 | 51.04 |
| SnAG [37] | 64.62 | 46.26 | 48.55 | 30.56 | - | - |
| UniVTG [27] | 58.01 | 35.65 | - | - | 58.86 | 40.86 |
| UniVTG w/PT | 60.19 | 38.55 | 42.41 | 21.55 | 65.43 | 50.06 |
| TimeSuite [62] | 67.10 | 43.00 | - | - | - | - |
| Mr.BLIP [34] | 69.31 | 49.29 | 53.92 | 35.55 | 74.77 | 60.51 |
| Qwen2-VL-7B [56] | 7.18 | 3.28 | 3.87 | 1.74 | - | - |
| Qwen2.5-VL-7B [2] | 60.32 | 34.27 | 16.96 | 8.75 | - | - |
| UniTime-SP | 74.33 | 53.71 | **54.81** | **36.62** | 77.76 | 63.29 |
| UniTime-Full | **75.27** | **56.85** | 53.67 | 35.90 | 76.72 | 62.65 |
| | (+5.07) | (+7.35) | (-0.25) | (+0.35) | (+1.95) | (+2.14) |

redundant video loading and computation, significantly accelerating training. To maintain consistency, we modify the attention mask to prevent cross-interaction between different query-answer pairs, and assign each query-answer sequence the same starting position index, directly following the video tokens' indices. For further implementation details, we refer readers to Appendix B.2.

## 3 Experiments

### 3.1 Experimental Setups

**Datasets.** To support universal temporal grounding, we compile a diverse dataset spanning varied scenes, genres, durations, and query types (*e.g.,* descriptions, questions, procedural instructions), as detailed in Table 1. For evaluation, we benchmark our method across three categories: (i) Short-video temporal grounding, including Charades-STA [51], ActivityNet-Captions [22], and QVHighlights [24]. (ii) Long-video temporal grounding, including Ego4D-NLQ [11] and TaCoS [47]. (iii) Video question answering (VideoQA), including two grounded benchmarks with temporally annotated queries (QaEgo4D [3], CG-Bench [5]) and two general benchmarks (MLVU [71], LongVideoBench [57]). The statistics of evaluation benchmarks we used are listed in Table 2.

**Evaluation Metrics.** For temporal grounding tasks, we adopt Recall@1 (R1) at multiple temporal intersection-over-union (IoU) thresholds and mean IoU (mIoU) as evaluation metrics. Specifically, we use IoU thresholds of 0.3 and 0.5 for long-video benchmarks, and 0.5 and 0.7 for short-video benchmarks. For general VideoQA tasks, we report standard accuracy metrics, while for grounded VideoQA tasks, we also evaluate R1, mIoU, and intersection-over-prediction (IoP). More details about metrics are in the Appendix A.1.

**Implementation Details.** Our framework is built on PyTorch, with Qwen2-VL-7B [56] as the base model. All experiments are conducted with a batch size of 8, using AdamW [32] optimizer with the learning rate 2e-4, and trained for one epoch with linear warmup during the first 3% of steps. The

Table 4: **Zero-shot performance on video temporal grounding benchmarks.** Zero indicates zero-shot. All models are evaluated in the zero-shot setting. Numbers in `gray` are sourced from the original paper; all others are tested by us using their released code and checkpoints. Improvements of UniTime-Zero over SoTA are highlighted in (green).

| Method | Ego4D-NLQ | | | TaCoS | | | Charades-STA | | | ANet-Captions | | | QVHighlights | | |
|---|---|---|---|---|---|---|---|---|---|---|---|---|---|---|---|
| | R1@.3 | R1@.5 | mIoU | R1@.3 | R1@.5 | mIoU | R1@.5 | R1@.7 | mIoU | R1@.5 | R1@.7 | mIoU | R1@.5 | R1@.7 | mIoU |
| UniVTG [27] | 6.48 | 3.48 | 4.63 | 5.17 | 1.27 | 4.40 | 25.22 | 10.03 | 27.12 | 11.10 | 4.06 | 16.86 | 14.13 | 4.42 | 23.38 |
| Mr.BLIP [34] | 6.49 | 3.20 | 5.37 | 24.59 | 14.32 | 17.94 | - | - | - | - | - | - | - | - | - |
| VTG-LLM [12] | 1.71 | 0.46 | 1.36 | 6.87 | 2.92 | 5.27 | 34.11 | 15.81 | 34.93 | 12.32 | 6.74 | 17.86 | 4.13 | 1.55 | 8.60 |
| Momentor [45] | - | - | - | - | - | - | 26.60 | 11.60 | 28.50 | **23.00** | 12.40 | **29.30** | - | - | - |
| VTimeLLM [18] | 1.67 | 0.77 | 2.52 | 9.30 | 3.90 | 8.08 | 34.30 | 14.70 | 34.60 | - | - | - | 26.13 | 11.16 | 29.42 |
| Timechat [48] | 1.67 | 0.79 | 1.30 | 3.77 | 1.60 | 2.95 | 29.73 | 12.53 | 31.20 | 16.38 | 8.36 | 21.53 | 8.32 | 4.26 | 14.25 |
| TimeMarker [6] | - | - | - | - | - | - | 51.90 | 26.90 | 48.40 | - | - | - | - | - | - |
| TimeSuite [62] | 0.88 | 0.43 | 0.94 | 6.75 | 2.50 | 5.71 | 48.95 | 24.65 | 45.91 | 16.56 | 9.28 | 22.03 | 12.32 | 9.16 | 21.30 |
| UniTime-Zero | **14.67** | **7.38** | **10.18** | **50.06** | **31.54** | **33.38** | **59.09** | **31.88** | **52.19** | 22.77 | **14.14** | 27.31 | **41.03** | **31.48** | **43.71** |
| | (+8.18) | (+3.90) | (+4.81) | (+25.47) | (+17.22) | (+15.44) | (+7.19) | (+4.98) | (+3.79) | (-0.23) | (+1.74) | (-1.99) | (+14.90) | (+20.32) | (+14.29) |

Table 5: **Closed-source Model Evaluation.** All models are evaluated on sampled subsets, with outputs lacking predicted timestamps excluded.

| Model | Ego4D-NLQ | | | Charades-STA | | |
|---|---|---|---|---|---|---|
| | R1@.3 | R1@.5 | mIoU | R1@.5 | R1@.7 | mIoU |
| Gemini-2.5-flash [8] | 15.29 | 15.29 | 13.93 | 36.46 | 13.54 | 38.72 |
| Gemini-2.5-pro [8] | **26.83** | **19.51** | **20.45** | 35.96 | 11.24 | 39.28 |
| GPT-4.1-mini [40] | 5.00 | 3.00 | 5.04 | 31.00 | 17.00 | 31.46 |
| GPT-4o [39] | 10.59 | 7.06 | 7.97 | 41.11 | 17.78 | 43.82 |
| Seed1.5-VL [54] | 19.39 | 10.20 | 13.74 | **90.82** | **63.27** | **73.69** |
| UniTime-Full | 25.00 | 17.00 | 17.20 | 81.00 | 61.00 | 70.11 |

Table 6: **Flexibility Verification.** Improvements over baseline are highlighted in (green). All UniTime variants use the data-specific setting.

| Method | Ego4D-NLQ | | | Charades-STA | | |
|---|---|---|---|---|---|---|
| | R1@.3 | R1@.5 | mIoU | R1@.5 | R1@.7 | mIoU |
| Qwen2-VL-2B [56] | 0.34 | 0.34 | 0.42 | 2.23 | 0.51 | 6.70 |
| +UniTime | 10.50 | 5.80 | 7.29 (+6.87) | 65.38 | 42.18 | 57.25 (+50.55) |
| Qwen2-VL-7B [56] | 0.48 | 0.17 | 0.46 | 7.18 | 3.28 | 8.98 |
| +UniTime | 24.79 | 16.83 | 17.25 (+16.79) | 74.33 | 53.71 | 63.15 (+54.17) |
| Qwen2.5-VL-7B [2] | 1.11 | 0.48 | 0.82 | 60.32 | 34.27 | 52.42 |
| +UniTime | 23.53 | 16.12 | 16.61 (+15.79) | 74.19 | 53.98 | 63.62 (+11.20) |
| InternVL2.5-2B [7] | 0.33 | 0.22 | 0.42 | 5.48 | 1.77 | 9.30 |
| +UniTime | 9.87 | 5.26 | 7.46 (+7.04) | 69.49 | 47.77 | 59.03 (+49.73) |
| InternVL2.5-8B [7] | 2.19 | 0.55 | 1.77 | 8.66 | 2.55 | 17.54 |
| +UniTime | 16.23 | 8.55 | 12.03 (+10.26) | 71.83 | 49.62 | 60.50 (+42.96) |

vision encoder is frozen, and the LLM is fine-tuned via LoRA [16] (rank = 8, alpha = 8). We sample frames at 2 fps, with $N_f^{\text{short}} = 128$ and $N_f^{\text{long}} = 1024$, capping input video at 16,384 tokens. The default segment length and replication factor are set to 32 and 4, respectively.

## 3.2 Comparison with State-of-the-arts on Video Temporal Grounding

**Dataset-specific Setting.** In Table 3, we compare our model against state-of-the-art methods on two long-video benchmarks and three short-video benchmarks. Under this setting, where models are trained and tested separately on each dataset, our method (**UniTime-SP**) outperforms all existing approaches (including traditional and MLLM-based models), achieving absolute improvements of 6.39% on Ego4D-NLQ, 9.10% on TaCoS, 5.45% on Charades-STA, 2.88% on ANet-Captions, and 2.89% on QVHighlights over prior best results.

**Universal Setting.** To assess broad applicability, we train a single model (**UniTime-Full**) on the largest collection of temporal grounding data (Table 1, Parts I & II). Unlike dataset-specific tuning, this setting prioritizes broad competence over specialized performance, ensuring the model robustly handles diverse video types and query styles in real-world scenarios. Without benchmark-specific fine-tuning, **UniTime-Full** achieves superior performance across all evaluated benchmarks, demonstrating its effectiveness as a universal solution for video temporal grounding.

**Zero-shot Setting.** To further test the generalization capability of our method, we pre-train a zero-shot model (**UniTime-Zero**) on Part I datasets in Table 1, while excluding all in-domain data (Part II). As shown in Table 4, our method achieves superior zero-shot performance, highlighting its remarkable generalization capability. The error analysis on ANet-Captions can be found in the Appendix C.2.

## 3.3 Comparison with Closed-source models on Video Temporal Grounding

To comprehensively assess UniTime, we benchmarked mainstream closed-source MLLMs to examine whether UniTime surpasses them or narrows the gap on VTG tasks. We conducted temporal grounding experiments on a randomly sampled subset from Ego4D and Charades. As shown in Table 5, Seed1.5-VL [54] attained the strongest performance on short videos (72.2 mIoU on Charades), while Gemini-2.5-Pro [8] exhibited superior grounding on long videos (20.5 mIoU on Ego4D). However, these closed-source models struggle to achieve high performance concurrently on both long and short

Table 7: **Ablation of the proposed modules.** Adaptive Scaling refers to adaptive frame scaling and Segment Retrieval indicates multi-granular temporal prediction.

| Adaptive Scaling | Multi-stage Inference | Segment Retrieval | Ego4D-NLQ R1@.3 | R1@.5 | mIoU |
|---|---|---|---|---|---|
| ✗ | ✗ | ✗ | 14.25 | 7.54 | 9.83 |
| ✓ | ✗ | ✗ | 14.00 | 7.51 | 9.83 |
| ✗ | ✓ | ✗ | 18.42 | 12.13 | 12.78 |
| ✓ | ✓ | ✗ | 17.91 | 11.99 | 12.39 |
| ✓ | ✓ | ✓ | **24.79** | **16.83** | **17.25** |

Table 8: **Robustness test on time shift and query decomposition.** QD indicates query decomposition, TS denotes time shift, and Ratio is TS performance relative to the original.

| Method | Charades-STA TS Shift | R1@.5 | R1@.7 | mIoU | Ratio | Charades-STA IoG of QD |
|---|---|---|---|---|---|---|
| VTimeLLM | ✗ | 28.53 | 12.35 | 32.00 | 53.67 | 68.99 |
|  | ✓ | 14.65 | 6.64 | 17.89 |  |  |
| Mr.BLIP | ✗ | 70.05 | 50.24 | 59.38 | 66.77 | 71.22 |
|  | ✓ | 51.32 | 26.91 | 43.63 |  |  |
| TimeSuite | ✗ | 48.65 | 24.01 | 45.32 | 66.26 | 47.07 |
|  | ✓ | 37.02 | 13.55 | 31.54 |  |  |
| UniTime-SP | ✗ | 73.36 | 53.74 | 62.87 | **80.20** | **74.88** |
|  | ✓ | 63.06 | 37.47 | 53.39 |  |  |

videos, indicating a lack of generalization. In contrast, UniTime achieves competitive performance compared to these closed-source models while maintaining more balanced capabilities for both short and long-form video grounding benchmarks, proving our approach to be overall the best in terms of consistent performance across varying video types.

### 3.4 Flexibility Verification

As discussed in Section 2.2, our approach is compatible with any MLLM that processes dynamic frame inputs. To evaluate flexibility and generalization, we conduct experiments across multiple baseline MLLMs (e.g., `Qwen2-VL`, `Qwen2.5-VL`, `InternVL2.5`) and model scales. As shown in Table 6, larger MLLMs outperform their smaller counterparts, and our method consistently improves temporal grounding performance regardless of the underlying architecture. These results demonstrate strong generalization and flexibility, and indicate that the effectiveness of our framework continues to benefit from advances in MLLM architectures and increased model scale, enabling further gains in temporal grounding as more powerful models become available.

### 3.5 Ablation Studies

**Effect of Individual Modules.** To assess the contribution of each module, we conduct ablation studies on the Ego4D-NLQ benchmark, with results presented in Table 7. The study examines three key components: "Adaptive Scaling" for video input processing; "Multi-stage Inference" implementing coarse-to-fine grounding for long videos; and "Segment Retrieval" for coarse temporal grounding through inserting timestamps before segments. The results demonstrate three key findings: (i) Adaptive frame scaling (row 2) maintains performance comparable to uniform sampling (row 1) when used independently. (ii) Multi-stage inference yields consistent improvements for temporal grounding (row 3) through progressive refinement from lower to higher frame rates, and for spatial grounding (row 4) via progressive refinement from lower to higher resolution, with both coarse-to-fine grounding strategies benefiting from this hierarchical approach. (iii) Most notably, incorporating multi-granular temporal prediction (row 5), where the model first retrieves a coarse relevant segment and then localizes the precise moment within it, yields substantial improvements across all metrics. These results highlight the effectiveness of the multi-stage temporal grounding framework. More details can be found in the Appendix A.2.1.

**Effect of Segment Length.** Our universal temporal grounding framework operates in two stages: (i) retrieving the video segment most relevant to the query; (ii) predicting the precise temporal interval within it. Segment length critically influences this process. We evaluate its impact using three metrics: Segment retrieval accuracy (R@1) measures the ability to identify the correct segment. Oracle grounding accuracy (oracle R1@0.3) reflects fine-grained localization given the ground-truth segment. Overall grounding accuracy (R1@0.3) assesses the overall end-to-end temporal grounding performance. As shown in Figure 3(a), longer segments improve segment retrieval accuracy but hinder fine-grained localization, resulting in lower oracle grounding performance. To balance this trade-off, we set the segment length to 32 for optimal overall accuracy.

**Effect of Replication Factor.** We investigate the impact of replication factor ($N_{rep}$), which balances long and short video samples during training. Figure 3(b) shows that increasing $N_{rep}$ initially improves segment retrieval but saturates and fluctuates later. In contrast, fine-grained grounding performance remains stable. Based on these results, we set $N_{rep} = 4$ for optimal performance.

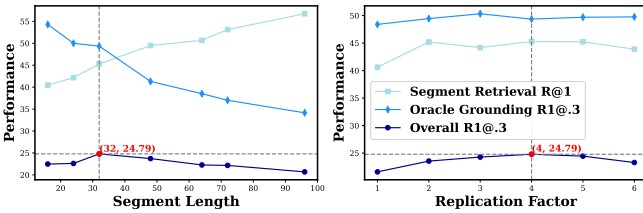 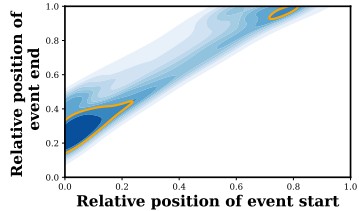

Figure 3: **Ablation on hyperparameters.** We decouple the effects of Segment Length (left) and Replication Factor (right) on temporal grounding performance.

Figure 4: **Temporal distribution bias in Charades-STA.** Darker colors indicate higher data density.

## 3.6 Robustness Test

Generative models are known to produce hallucinations in their outputs [28]. To evaluate robustness, we conduct experiments on the Charades-STA benchmark from two perspectives: (i) perturbing event positions to assess robustness against temporal distributional bias, and (ii) decomposing complex queries into simpler object-based questions to test the model's reliability across query types.

**Time Shift on Event.** Previous studies [14, 41] reveal significant biases in event distributions within temporal grounding datasets. As seen in Figure 4, annotated events in Charades-STA are heavily concentrated near the start or end of videos. To assess robustness against such bias, we perturb the data distribution by sampling clips that contain the event segment but place it at randomized positions. As shown in Table 8, our method exhibits greater robustness than other MLLM-based approaches, as evidenced by the higher performance ratio relative to the unperturbed baseline.

**On Decomposition of Query.** Accurate localization of complex queries fundamentally depends on understanding the referenced objects or events. To address this, we prompt Qwen2 [59] to decompose each Charades-STA query into a set of object-based questions in the form of "When does `<object>` appear?", where `<object>` denotes an entity that can be grounded in the original query. For each decomposed query, we compute the intersection-over-ground-truth (IoG) between its predicted segment and the ground-truth segment. It is worth noting that this is only an approximate estimation, as an entity may appear in multiple temporal segments, while annotations typically provide only a single segment. As shown in Table 8, our approach demonstrates superior query understanding and more reliable localization, with fewer hallucinations compared to other methods.

Recently, a related work [21] focuses on prediction consistency: they also curate test data by rephrasing original queries and shifting ground-truth moments in videos to probe grounding consistency. For detailed tests and comparisons, please refer to the Appendix D.

## 3.7 Downstream Task

To validate the effectiveness of our universal temporal grounding model in facilitating long video understanding as discussed in Section 1, we evaluate on four VideoQA benchmarks. As a baseline, we uniformly sample 32 frames from each video and input them to the `Qwen2-VL-7B` model for answer generation. To assess the impact of different temporal grounding approaches on downstream VideoQA, we first use each model to predict the relevant temporal segment for each question, then uniformly sample 32 frames from the predicted segment and feed them into `Qwen2-VL-7B` for answer prediction. As shown in Table 9, our method exhibits superior generalization compared to existing temporal grounding models, demonstrating robust grounding performance on out-of-domain data and enhancing long video understanding. Refer to Appendix A.4 for VideoQA evaluation details.

Table 9: **Performance on VideoQA benchmarks.** R@IoU is the mean Recall@1 at IoU thresholds $[0.1 : 0.1 : 0.5]$. For grounded VideoQA, IoP and mIoU are used as grounding metrics.

| Method | QaEgo4D | | | | CG-Bench | | | | MLVU | LongVideoBench |
| --- | --- | --- | --- | --- | --- | --- | --- | --- | --- | --- |
| | IoP | R1@.5 | mIoU | Acc. | IoP | R@IoU | mIoU | Acc. | Acc. | Acc. |
| Uniform Sample | 3.67 | - | - | 49.60 | 1.08 | - | - | 33.87 | 60.53 | 54.82 |
| UniVTG [27] | 9.80 | 6.20 | 7.76 | 50.00 | 1.37 | 4.92 | 3.86 | 34.87 | 62.56 | 54.67 |
| VTimeLLM [18] | 3.87 | 1.40 | 3.68 | 48.60 | 1.62 | 1.22 | 1.58 | 34.60 | 59.52 | 54.30 |
| TimeSuite [62] | 6.28 | 0.20 | 1.22 | 48.80 | 5.06 | 2.89 | 2.09 | 32.47 | 58.51 | 53.25 |
| UniTime-Full | **25.29** | **19.00** | **18.44** | **55.51** | **16.64** | **17.66** | **11.63** | **40.30** | **66.50** | **56.47** |

# 4    Related Work

**Video Temporal Grounding (VTG)** tasks can typically be categorized into short- and long-duration scenarios. For short videos, SOTA methods primarily adopt DETR-like architectures [4, 24, 36, 35, 10]. For example, Moment-DETR [24] formulates moment retrieval as a set prediction problem with a transformer-based encoder–decoder, enabling end-to-end prediction of moment coordinates and saliency. Recent works further enhance performance by introducing prior knowledge [20] and advanced attention mechanisms [36, 35]. Non-DETR methods, such as UMT [29] and Mr.BLIP [34], leverage multi-modal cues and large language models, respectively. While effective for short videos, these methods struggle with long videos, where sparse relevant moments create a "needle-in-a-haystack" challenge. Recent works [15, 43] employ coarse-to-fine alignment and hierarchical pipelines to address this. Despite the progress on individual benchmarks, fundamental differences between short and long videos hinder the development of unified models. While UniVTG [27] attempts to bridge this gap, its lightweight architecture limits generalization, especially in out-of-domain scenarios [50]. This motivates our work to develop a universal grounding model capable of (i) handling videos of arbitrary duration, (ii) robustly generalizing across domains, and (iii) excelling in downstream tasks like video question answering.

**Temporal Grounding with Multi-modal Language Models (MLLMs).** While MLLMs have achieved remarkable success across various multi-modal tasks [23, 44, 31], accurate temporal grounding remains challenging. Existing temporal integration approaches in MLLMs can be grouped into three paradigms: (i) time-agnostic models: VTimeLLM [18] processes fixed-length videos (100 frames) without explicit temporal cues, predicting normalized moment positions; LITA [19] replaces the predicted textual timestamps with special time tokens. The lack of explicit temporal signals often leads to suboptimal performance. (ii) implicit timestamp-encoded models: These methods incorporate timestamp-aware encoders, such as TimeChat [48] and TimeSuite [62], which fuse frame and temporal embeddings. VTG-LLM [12] uses absolute time embeddings, while Qwen2.5-VL [2] employs MRoPE for temporal modeling. Despite advances, these methods require extensive pretraining and may hallucinate timestamps due to implicit temporal representations. (iii) explicit temporal marking models: Methods like Mr.BLIP [34], TimeMarker [6], and VideoLLaMA3 [63] prepend textual timestamps to video frames, leveraging retrieval capabilities of MLLMs to match DETR-based performance. Although these approaches have advanced short video temporal grounding, they are constrained by context windows and GPU memory, limiting scalability to long videos. To address this, we propose a framework that adaptively adjusts scales of video frames, combined with a multi-scale, coarse-to-fine iterative temporal grounding strategy tailored for long video understanding.

**Long Video Understanding** is significantly more challenging than short video or image-based tasks due to complex temporal dynamics and substantial redundancy. The large number of frames dramatically increases memory and computational demands, making dense sampling impractical. Existing video MLLMs [2, 56, 69] often rely on uniform frame sampling, which risks omitting critical information. To address this, some approaches compress frames into minimal token sets to reduce computational costs [49, 26, 67], while others focus on more effective frame sampling strategies [60, 52, 17, 9]. Despite these advancements, temporal grounding—crucial for retrieving salient events—remains underexplored in the context of long videos. In this work, we leverage temporal grounding to identify key events, enhancing the understanding of long-form video content.

# 5    Conclusion

In conclusion, this paper presents **UniTime**, a universal video temporal grounding model based on MLLMs. At its core, UniTime interleaves timestamp tokens with video tokens and retrieves relevant timestamps for natural language queries, thereby enabling precise temporal localization. By employing adaptive frame scaling, UniTime constructs inputs with varying spatial granularities for videos of different durations. By further inserting timestamps at multiple temporal granularities, the model supports multi-scale temporal prediction. As a result, it can perform coarse-to-fine temporal grounding, first coarsely localizing segments in long videos and then refining predictions to fine-grained windows in short videos. To further enhance efficiency, we propose a video-centric training paradigm that reduces redundant computation. Extensive experiments on multiple benchmarks demonstrate that UniTime, pre-trained on large-scale datasets without the need for dataset-specific finetuning, consistently outperforms existing methods in both temporal grounding and downstream long-video question answering tasks.

## Acknowledgements

This work is supported by National Key R&D Program of China (No.2022ZD0161400).

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

## Appendix / supplemental material

In the appendix, we provide additional experimental details (Appendix A), implementation details (Appendix B), qualitative results (Appendix C), grounding consistency evaluation (Appendix D), more ablation studies (Appendix E), discussions of limitations and future work (Appendix F), and broader impacts (Appendix G).

## A  Experimental Details

In this section, we provide additional experimental details, including benchmark details (Appendix A.1), ablation details (Appendix A.2), robustness test details (Appendix A.3)and downsteam task details (Appendix A.4).

### A.1  Benchmark Details

To evaluate the performance of our model on temporal grounding and VideoQA tasks, we employ three key metrics: Intersection-over-Union (IoU), Intersection-over-Prediction (IoP), and Intersection-over-Ground Truth (IoG). These metrics quantify the alignment between the predicted temporal window $T^{\text{pred}}$ and the ground truth temporal window $T^{\text{gt}}$. The formal definitions are as follows:

$$\text{IoU} = \frac{|T^{\text{pred}} \cap T^{\text{gt}}|}{|T^{\text{pred}} \cup T^{\text{gt}}|}, \quad \text{IoP} = \frac{|T^{\text{pred}} \cap T^{\text{gt}}|}{|T^{\text{pred}}|}, \quad \text{IoG} = \frac{|T^{\text{pred}} \cap T^{\text{gt}}|}{|T^{\text{gt}}|}.$$

Here, $|T^{\text{pred}} \cap T^{\text{gt}}|$ denotes the duration of the overlapping region between the predicted window $T^{\text{pred}}$ and the ground truth window $T^{\text{gt}}$. $|T^{\text{pred}} \cup T^{\text{gt}}|$ represents the duration of their union, while $|T^{\text{pred}}|$ and $|T^{\text{gt}}|$ denote the durations of the predicted and ground truth windows, respectively.

IoU measures the overall alignment between $T^{\text{pred}}$ and $T^{\text{gt}}$, providing a balanced assessment of both precision and recall by considering the overlap relative to their union. IoP emphasizes precision, indicating the proportion of the predicted window that overlaps with the ground truth. Conversely, IoG focuses on recall, reflecting the proportion of the ground truth window that is covered by the prediction. Collectively, these metrics offer a comprehensive evaluation of the model's ability to accurately localize temporal windows with respect to the ground truth.

### A.2  Ablation Details

#### A.2.1  Effect of Individual Modules

As shown in Table 7, we systematically evaluate the effect of each individual module. The detailed settings are as follows: (i) **Row 1**: We uniformly sample 32 frames from each video and obtain the corresponding timestamps. These frames and timestamps are then fed into UniTime for temporal grounding. (ii) **Row 2**: We sample the video at 2 fps and apply frame scaling. The sampled frames and their timestamps are input into UniTime for temporal grounding. (iii) **Row 3**: On top of uniform sampling, we introduce multi-stage inference. Specifically, after obtaining an initial prediction using the uniformly sampled frames (as in row 1), we perform a second round of uniform sampling within the predicted interval and feed these frames into the model for refined localization. (iv) **Row 4**: Similarly, we apply multi-stage inference to the adaptive frame scaling. After the initial prediction (as in row 2), we resample at 2 fps and apply frame scaling within the predicted interval, then input these frames into the model for further refinement. (v) **Row 5**: We adopt the coarse-to-fine temporal grounding strategy described in Section 2.2. Unlike row 4, where the model predicts fine-grained intervals in the first round due to densely sampled timestamps, row 5 partitions the video into coarse segments and only requires the model to predict the relevant segment in the first round. In the second round, temporal grounding is performed within the selected segment to localize the precise moment.

#### A.2.2  Effect of Segment Length and Replication Factor

As shown in Figure 3, we decouple the process of coarse-to-fine temporal grounding to analyze the effect of segment length and replication factor. Below, we provide detailed descriptions of the capabilities of each component after decoupling.

Overall, the coarse-to-fine temporal grounding process can be divided into the following two stages. First, we retrieve candidate segments from the video as follows:

$$\{\mathbf{S}_r\}_{r=1}^{N_r} = \Phi_{\text{UniTime}}(\mathcal{V}, \mathcal{T}, \mathcal{Q}) = \Phi_{\text{UniTime}}([\mathbf{T}_1; \mathbf{S}_1; \mathbf{T}_2; \mathbf{S}_2; \cdots ; \mathbf{T}_{N_s}; \mathbf{S}_{N_s}; \mathcal{Q}])$$

where $\mathbf{S}_r$ denotes the retrieved segments and $N_r$ is the number of retrieved segments. Next, we perform fine-grained localization within the retrieved segments:

$$\mathcal{A} = \{(s_1, e_1), \ldots, (s_K, e_K)\} = \Phi_{\text{UniTime}}(V|_{\{\mathbf{S}_r\}_{r=1}^{N_r}}, \mathcal{T}, \mathcal{Q})$$

where each $s_k, e_k \in \mathcal{T}$ represents the predicted start and end timestamps, respectively. Then, the evaluation metrics used in Section 3.5 and Figure 3 are defined as follows:

Segment retrieval accuracy (R@1) indicates the accuracy of segment retrieval, i.e., the proportion of cases where any predicted segment $\mathbf{S}_r^{\text{pred}}$ is contained within the set of ground-truth segments $\{\mathbf{S}_r^{\text{gt}}\}$. Oracle grounding accuracy (oracle R1@0.3) reflects the localization performance when the model is provided with the ground-truth segment as input, i.e., $\mathcal{A}^{\text{oracle}} = \Phi_{\text{UniTime}}(V|_{\{\mathbf{S}_r^{\text{gt}}\}}, \mathcal{T}, \mathcal{Q})$. Overall grounding accuracy (R1@0.3) measures the end-to-end temporal grounding performance, where $\mathcal{A}^{\text{pred}} = \Phi_{\text{UniTime}}(V|_{\{\mathbf{S}_r^{\text{pred}}\}}, \mathcal{T}, \mathcal{Q})$.

### A.3 Robustness Test Details

### A.3.1 Time Shift on Event

The temporal distribution bias in Charades-STA is illustrated in Figure 4. In addition, we also conduct statistical analyses on TACoS, ANet-Captions, and QVHighlights, with the results presented in Figure 5. Significant temporal distribution bias is also observed in TACoS and ANet-Captions, whereas QVHighlights does not exhibit such bias.

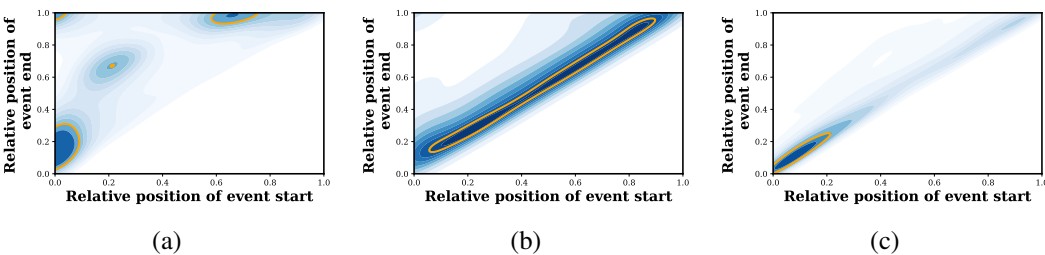

(a)          (b)          (c)

Figure 5: Temporal distribution bias in (a) ANet-Captions. (b) QVHighlights. (c) TACoS. Darker colors indicate higher data density. , and the yellow curve outlines the region of greatest density.

### A.3.2 Decomposition of Query

The prompts provided to Qwen2 [59] for decomposing each Charades-STA query into a set of object-centric questions are as follows:

```
System:
You are Qwen, created by Alibaba Cloud. You are a helpful assistant.
System:
Analyze the given query and:
1. Identify ONLY concrete, specific objects (nouns) that are:
   - Tangible physical items
   - Clearly named (not pronouns/ambiguous)
2. STRICTLY EXCLUDE:
   - All human references (person, he, she, they, etc.)
   - Ambiguous terms (something, anything, things, etc.)
   - Pronouns (it, they, them)
   - Abstract concepts
3. For each valid object, generate EXACTLY ONE question:
   "When does [OBJECT] appear?"

Negative Examples (BAD):
```

```
Input: "Someone left some items on the furniture"
Wrong Output:
-When does the furniture appear?
-When does some items appear? # <-- AMBIGUOUS TERM SHOULD BE EXCLUDED

Positive Examples (GOOD):
Input: "The machine processed the raw materials during the night"
Output:
-When does the machine appear?
-When does the raw materials appear?
-When does the night appear?
User:
Analyze: <query>
```

## A.4 Downstream Task Details

### A.4.1 VideoQA with Temporal Grounding

We use `Qwen2-VL-7B` as the VideoQA model for answer generation. By default, it processes long videos by uniformly sampling 32 frames. However, this sampling strategy may lead to the omission of critical information. To investigate whether temporal grounding models can compensate for this issue, we adopt the following procedure. First, we use different video temporal grounding models to localize the relevant segments for each question. Then, we crop the localized video intervals and input them into `Qwen2-VL-7B`. Specifically, for cropped video segments shorter than 32 seconds, we extend their duration from the center to 32 seconds. Within each interval, we again uniformly sample 32 frames for answer generation.

### A.4.2 Prompt template for VideoQA

We use the same prompt template for all multiple-choice VideoQA benchmarks:

```
System:
You are a helpful assistant.
User:
<video>
Question: <question>
Options:
(A) <Option_A>
(B) <Option_B>
(C) <Option_C>
(D) <Option_D>
Please only give the best option.
Best Option:
Assistant:
```

## B Additional Implementation Details

### B.1 Prompt template for UniTime

The prompt template used for coarse-grained segment retrieval as follows:

```
System:
You are a helpful assistant.
User:
<timestamps & frames>
This is a sequence interleaved with timestamps and frames.
Your task is to identify the specific timestamp(s) when the given query appears.
Query: <Query>
Answer:
Assistant:
```

The prompt template used for fine-grained temporal grounding as follows:

```
System:
You are a helpful assistant.
User:
<timestamps & frames>
This is a sequence interleaved with timestamps and frames.
Your task is to identify the temporal window (start and end timestamps) when the
    given query appears.
Query: <Query>
Answer:
Assistant:
```

## B.2 Video-centric Training

In video-centric training, we first sample a video from the dataset and gather all associated queries with their corresponding answers. These queries and answers are concatenated and paired with the video as input, forming a sequence such as

$$\left[v_1, v_2, \ldots, v_N, Q_1, A_1, Q_2, A_2, \ldots, Q_{N_{\text{sample}}}, A_{N_{\text{sample}}}\right]$$

. As illustrated in Figure 6, we apply attention masks to prevent different queries and answers from attending to others. Additionally, each query-answer sequence is assigned the same starting Rotary Position Embedding index, immediately following the indices of the video tokens.

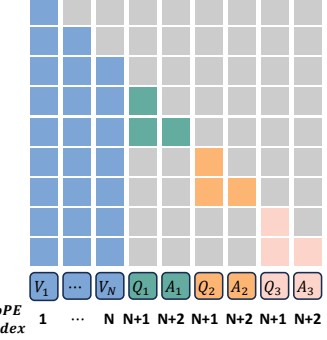

Figure 6: Illustration of the video-centric training paradigm.

## C Qualitative Results

### C.1 Qualitative Results for Ego4D-NLQ

In Figure 7, we present qualitative results of UniTime on Ego4D-NLQ. In the success cases, UniTime demonstrates precise coarse-grained segment retrieval and fine-grained temporal grounding capabilities. Despite the limited variation in video scenes, which are primarily composed of fine-grained interactions between hands and objects, the model consistently demonstrates robust performance. In the failure cases, although the model retrieves the correct segment, it fails in fine-grained localization. As shown in the specific localization frames, this error occurs because the object referred to in the query appears multiple times in the video, while the ground truth only includes one of these occurrences.

### C.2 Qualitative Results for ANet-Captions

As shown in Table 2, UniTime achieves substantial performance gains over competing methods across all benchmarks, except for ANet-Captions. We analyze this discrepancy below, with Figure 8 visualizing UniTime's predictions on ANet-Captions.

In successful cases (Figure 8(a)), UniTime achieves precise temporal grounding. Despite the limited diversity of video scenes, which predominantly feature fine-grained interactions between hands and objects, the model consistently exhibits robust performance.

However, our analysis of the failure cases indicates that most can be traced to limitations in ANet-Captions's annotation scheme. As illustrated in Figure 8(b), multiple occurrences of the same event are often annotated with only a single temporal interval. This occurs because captions are ordered sequentially, causing annotators to overlook recurring events.

Additionally, many failures arise from incorrect labels, as shown in Figure 8(c).

## D Grounding Consistency Evaluation

To further evaluate UniTime's robustness in temporal grounding, we conduct grounding consistency assessment on Charades-CON and ActivityNet-CON [21], which are curated from subsets

**Query:** Where did I throw the trash from the plate?

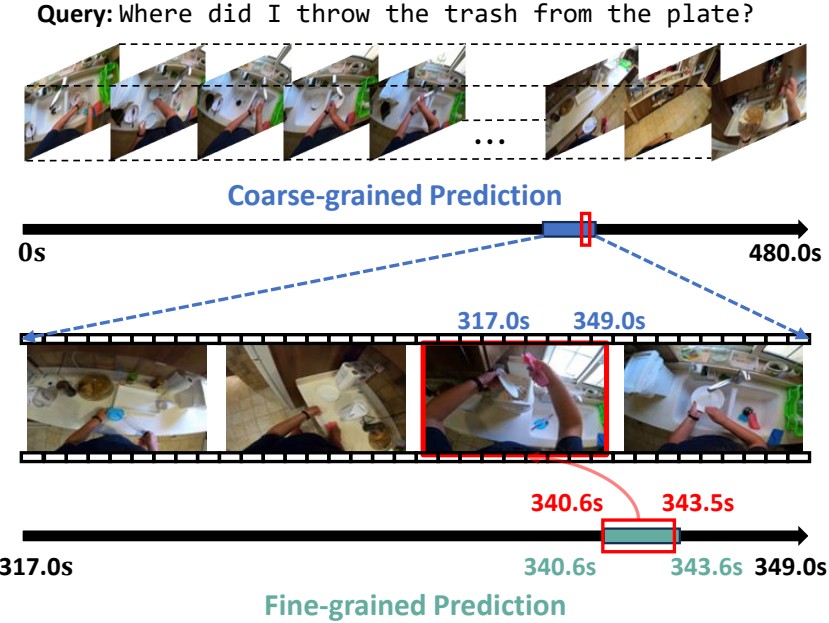

(a) success case

**Query:** Where is the paper towel?

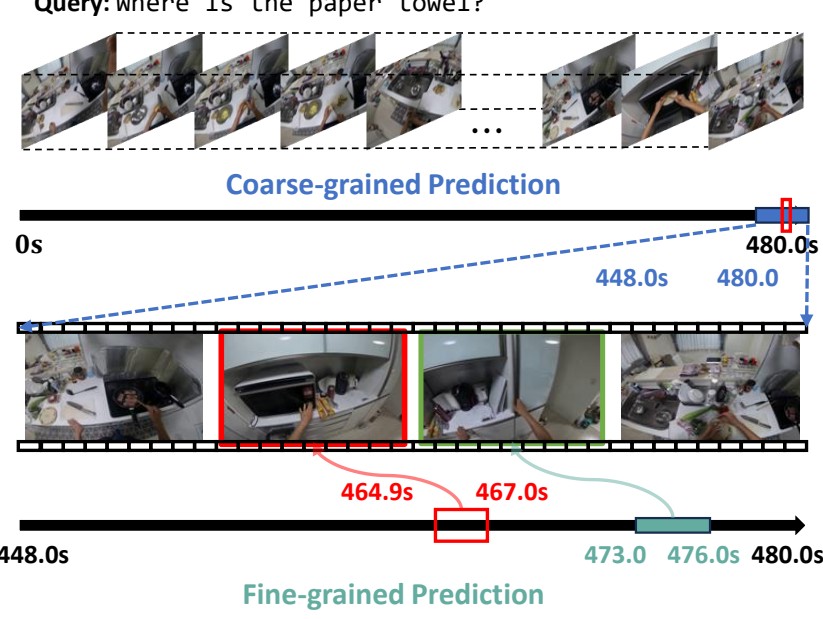

(b) failure case

Figure 7: Qualitative Results for long-video temporal grounding on the Ego4D-NLQ benchmark. **Red** indicates the ground truth (GT), **blue** denotes the results of coarse-grained segment retrieval, and **green** represents the results of fine-grained temporal grounding.

**Query:** A woman rolls up a towel.

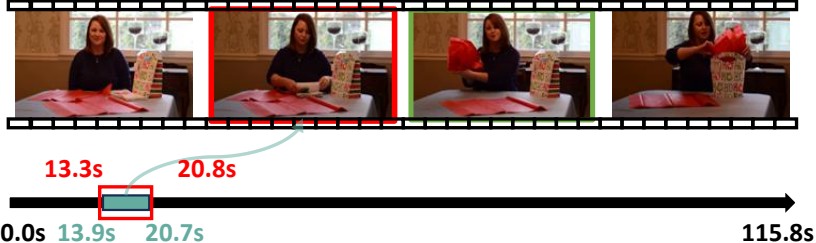

(a) Success case.

**Query:** A man starts break dancing on the floor.

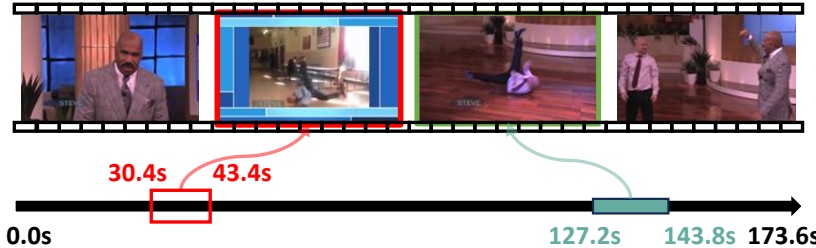

(b) Failure case resulting from incomplete annotations of multi-hop events.

**Query:** The woman then lights a candle.

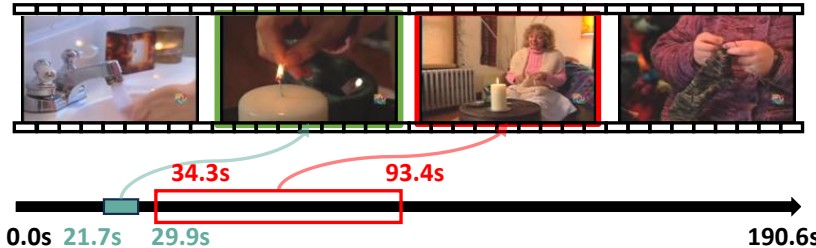

(c) Failure case caused by incorrect labels.

Figure 8: Qualitative Results for short-video temporal grounding on the ANet-Captions benchmark. **Red** indicates the ground truth (GT), **blue** denotes the results of coarse-grained segment retrieval, and **green** represents the results of fine-grained temporal grounding.

Table 10: **Grounding consistency evaluation of Video-MLLMs.** Relative consistency scores are in brackets.

| Method | Charades-CON | | | ActivityNet-CON | | |
|---|---|---|---|---|---|---|
| | Ground | R-Ground | S-Ground | Ground | R-Ground | S-Ground |
| GPT-4o [39] | 28.5 | 21.2 (74.3) | 9.3 (32.8) | 26.8 | 18.1 (67.5) | 10.4 (38.8) |
| Gemini 1.5 Flash [55] | 34.6 | 29.7 (85.7) | 24.8 (71.7) | 37.8 | **30.8 (81.4)** | **24.8 (65.6)** |
| Video-LLaMA [64] | 14.2 | 10.6 (74.9) | 5.3 (37.6) | 12.8 | 8.5 (66.8) | 7.2 (56.8) |
| Video-LLaMA + VTune [21] | 54.4 | 38.2 (70.3) | 10.9 (20.0) | 33.0 | 24.7 (74.8) | 10.0 (30.2) |
| TimeChat [48] | 30.5 | 25.0 (82.1) | 5.6 (18.5) | 4.6 | 2.9 (64.1) | 1.0 (21.2) |
| TimeChat + VTune [21] | 76.2 | 69.2 (**90.8**) | 36.2 (47.5) | 37.4 | 28.3 (75.6) | 10.6 (28.3) |
| UniTime-Full | **81.5** | **73.9** (90.7) | **58.3 (71.6)** | **49.1** | 30.2 (61.5) | 21.3 (43.4) |

of CharadesSTA and ActivityNet-Captions, respectively. The grounding consistency targets two complementary consistency dimensions:

i) Rephrased Grounding (R-Ground). This metric evaluates whether a model's moment predictions remain consistent across a query $q$ and its aligned rephrased variant $\hat{q}$. Concretely, the IoU between $m = \text{Temp}_G(q)$ and $\hat{m} = \text{Temp}_G(\hat{q})$ is computed. For each original query, the average IoU over three aligned variants is reported, reflecting the alignment consistency of predicted moments under paraphrasing.

ii) Shifted Grounding (S-Ground). This metric assesses whether a model consistently grounds the same visual content when that content is shifted to different temporal positions. Specifically, while preserving the internal frame sequence of the ground-truth moment $m_{\text{gt}}$, it is randomly shifted to a new moment $m_s$, with evaluations conducted on the resulting predictions.

Experimental results, summarized in the Table 10, demonstrate that UniTime not only exhibits strong temporal grounding performance (Ground) but also achieves excellent consistency and robustness (R-Ground and S-Ground). UniTime attains performance comparable to VTune [21], which extends instruction tuning with explicit consistency objectives, and in some cases surpasses it.

# E More Ablation Studies

## E.1 Effect of Video Processing Strategies

To justify our design choice in Section 2.2, i.e., resizing short videos and compressing long videos, we conduct an ablation study on the Ego4D-NLQ dataset. This study evaluates how different video processing strategies impact both grounding performance and computational efficiency.

Table 11: **Ablation on video processing strategies.** Seg Retrieval Acc. indicates the accuracy of coarse-grained segment retrieval.

| Short Video | Long Video | R1@.3 | R1@.5 | mIoU | Seg Retrieval Acc. | Training Time |
|---|---|---|---|---|---|---|
| Frame Resizing | Token Compression | 24.79 | 16.83 | 17.25 | 45.28 | 21 h |
| Frame Resizing | Frame Resizing | 18.53 | 12.67 | 12.88 | 35.36 | 12 h |
| Token Compression | Token Compression | 24.06 | 16.18 | 16.71 | 45.28 | 24.5 h |

Our findings indicate that: (i) For long videos, adaptive frame scaling reduces tokens per frame while keeping the frame rate fixed. Though some visual cues are lost, this feature-level token compression retains more semantic information than frame resizing, which discards input resolution. This leads to better segment retrieval accuracy and recall (row 1 vs. row 2). (ii) For short clips with sufficient tokens, both methods perform similarly (row 1 vs. row 3). However, token compression requires high-resolution inputs before merging, incurring higher computational costs. In UniTime-Full, using token compression for short videos increased training time by 5 days compared to frame resizing.

Therefore, rather than employing a unified strategy, we adopt a hybrid approach to balance performance and efficiency: for short videos, we apply frame resizing to reduce computational overhead, while for long videos, we leverage token compression to better preserve visual details. This design choice enables our method to achieve strong performance with practical training costs.

### E.2 Comparison of Temporal Information Encoding

We conduct ablation experiments on Charades-STA to compare the effectiveness of Timestamp Token Insertion versus Dense Position Encodings.

Table 12: **Ablation on temporal information encoding.**

| Method | R1@.5 | R1@.7 | mIoU |
|---|---|---|---|
| Dense Position Encodings | 48.44 | 27.15 | 44.85 |
| Timestamp Token Insertion (UniTime-SP) | 74.33 | 53.71 | 63.15 |

Our base model, `Qwen2-VL-7B`, utilizes Multimodal Rotary Position Embedding (MRoPE), which decomposes position embeddings into temporal, height, and width components. After fine-tuning, temporal grounding remains suboptimal with dense position encodings. In contrast, timestamp token insertion yields significant gains. We attribute this improvement to the MLLM's strong multimodal retrieval ability and the simplification of the task into (i) identifying question-relevant video segments and (ii) directly reading their corresponding timestamp tokens.

### E.3 Ablation Study of Adaptive Frame Scaling Hyperparameters

In this section, we conduct an ablation study on the threshold hyperparameters $N_f^{\text{long}}$, $N_f^{\text{short}}$, and the token budget $N_{\text{total}}$ within Adaptive Frame Scaling, elucidating the rationale behind our parameter choices.

**Threshold $N_f^{long}$.** This parameter specifies the maximum video length that can be processed in a single pass. For videos with length $\leq N_f^{\text{long}}$, we process the input directly; for videos with length $> N_f^{\text{long}}$, we partition the input into segments, process each segment independently, and then aggregate the predictions. As shown in Table 13, varying $N_f^{\text{long}}$ has only a marginal effect on accuracy because over-length inputs are segmented and subsequently fused. Moreover, Adaptive Frame Scaling maintains a fixed token budget per segment, so reducing $N_f^{\text{long}}$ increases the number of segments and the total token count at inference, thereby incurring higher latency. Balancing efficiency and accuracy, we set $N_f^{\text{long}} = 1024$.

**Threshold $N_f^{short}$.** This hyperparameter determines whether inference is executed in a single stage or via a multi-stage (coarse-to-fine) pipeline. For inputs with temporal length $\leq N_f^{\text{short}}$, we use single-stage processing; for inputs with temporal length $> N_f^{\text{short}}$, we employ multi-stage processing. As shown in Table 14, within the 64–128 s duration range, the performance difference between multi-stage ($N_f^{\text{short}}$=64) and single-stage ($N_f^{\text{short}} \geq 128$) configurations is modest. However, a lower threshold increases average inference latency by routing more inputs to the multi-stage pipeline. For inputs in the 128–256 s range, multi-stage processing yields substantially higher accuracy (54.57 vs. 42.90 mIoU). To balance accuracy and efficiency, we set $N_f^{\text{short}}$=128.

**Token budget $N_{\text{total}}$.** This parameter specifies the maximum number of tokens permitted. As shown in Table 15, performance improves monotonically as the token budget increases, up to the available GPU capacity. Accordingly, we set $N_{\text{total}} = 16,384$, which corresponds to the upper limit supported by our hardware.

## F  Limitations & Future Work

While UniTime demonstrates exceptional performance on various video temporal grounding and video QA benchmarks, it still has several limitations that warrant further exploration: (i) UniTime is currently constrained to temporal grounding tasks. To enable broader applications in MLLMs, it requires more diverse training data with dense temporal annotations. Incorporating such data into the pretraining process of MLLMs could unlock their potential for handling more temporally complex tasks, such as dense video captioning. (ii) Although UniTime enhances MLLMs with temporal grounding capabilities, relying solely on temporal grounding data limits their reasoning and

Table 13: **Ablation of the threshold $N_f^{\text{long}}$ on Ego4D.**

| $N_f^{long}$ | R1@.3 | R1@.5 | mIoU |
|---|---|---|---|
| 1024 | 24.79 | 16.83 | 17.25 |
| 512 | 24.72 | 16.63 | 16.95 |
| 256 | 23.94 | 16.10 | 16.68 |

Table 14: **Ablation of the threshold $N_f^{\text{short}}$ on TaCoS.**

| $N_f^{short}$ | mIoU (64-128s) | mIoU (128-256s) | mIoU Avg. |
|---|---|---|---|
| 64 | 58.43 | 54.57 | 50.28 |
| 128 | 56.59 | 54.57 | 50.00 |
| 256 | 56.59 | 42.90 | 46.82 |

Table 15: **Ablation of the token budget $N_{\text{total}}$ on Ego4D.**

| Budget | R1@.3 | R1@.5 | mIoU |
|---|---|---|---|
| 16384 | 24.79 | 16.83 | 17.25 |
| 8192 | 24.08 | 16.10 | 16.69 |
| 4096 | 23.04 | 15.07 | 15.96 |

question-answering abilities. The ultimate objective is to develop MLLMs that seamlessly integrate localization, reasoning, and question-answering into a unified framework. (iii) The current approach to processing long videos in UniTime relies on a fixed segment length, which lacks flexibility. Future work could investigate adaptive segment lengths based on the information density of videos, enabling more efficient and context-aware video processing.

## G    Broader Impacts Statement

Our research advances the field of video understanding by developing more accurate and efficient models for temporal grounding and video question answering. While temporal grounding with UniTime offers significant benefits for a variety of video understanding applications, unexpected behaviors may occasionally result in misrepresentations of the underlying video content. For applications that demand extremely precise temporal localization for safety-critical decisions, such as embodied intelligence tasks, it is crucial to carefully manage any such behaviors. To ensure the reliability of systems that rely on temporal grounding predictions, we recommend conducting thorough evaluations and implementing robust mitigation strategies to minimize potential risks. Through these precautions, the overall safety and effectiveness of these applications can be substantially enhanced.

