# OpenReview forum: "Universal Video Temporal Grounding with Generative Multi-modal Large Language Models"
_NeurIPS.cc/2025/Conference — NeurIPS 2025 poster_

### Official Review · Reviewer_F2GW · 2025-06-24

**Clarity:** 3
**Significance:** 3
**Originality:** 2
**Rating:** 4
**Confidence:** 4

**Summary:**

The paper presents UniTime, a framework for universal video temporal grounding using generative multimodal large language models. The approach enables precise localization of relevant video segments given natural language queries. To support long and short videos, the model introduces timestamp token interleaving, adaptive frame scaling, and a coarse-to-fine inference strategy. Experiments show strong results across five video grounding and four video question answering benchmarks, including zero-shot and fine-tuned settings.

**Questions:**

1. How were the thresholds and token budgets in adaptive frame scaling determined, and how sensitive is performance to these choices?

2. Does relying solely on LLM-decoded textual timestamps limit fine-grained precision, and could auxiliary supervision improve it?

3. What is the computational overhead of the multi-stage inference pipeline compared to one-stage baselines?

4. How does the model handle ambiguous queries with multiple valid grounding regions or misleading timestamps?

5. Has partial or full fine-tuning of the visual encoder been explored, and would it help under domain shift?

**Ethical Concerns:**

["NO or VERY MINOR ethics concerns only"]

**Final Justification:**

I believe the paper has merit in its current form, and while there may be areas for improvement, I maintain my score based on its overall contribution.

**Limitations:**

yes

**Paper Formatting Concerns:**

No issues observed.

**Quality:**

2

**Strengths And Weaknesses:**

**Strengths:**

1. The paper aims to address an important and challenging problem of temporal grounding in both short and long videos using a unified framework.

2. The writing is clear and well-structured, with detailed descriptions of the method, training strategy, and experimental setup.


**Weakness:**

1. The temporal reasoning mechanism relies entirely on textual timestamp tokens and LLM decoding, which limits its temporal resolution and introduces potential inconsistencies in fine-grained prediction, especially for closely adjacent events.

2. The adaptive frame scaling policy is based on fixed token budget allocation heuristics and does not incorporate content-aware or query-guided frame importance estimation, which could lead to suboptimal grounding in visually dense regions.

3. The use of interleaved timestamp tokens assumes that LLMs can resolve temporal positions through text-level retrieval, but the model lacks explicit temporal alignment supervision or contrastive objectives to ensure accurate timestamp grounding.

4. The training strategy employs video-centric batching with LoRA fine-tuning on frozen visual backbones, but does not explore joint optimization of vision-language alignment, which may restrict performance in out-of-domain or visually ambiguous cases.

5. The multi-stage inference pipeline improves grounding granularity but increases latency and computational cost, and the paper does not quantify this overhead or compare against single-stage efficient alternatives.

---

> ### Author Rebuttal · Authors · 2025-07-31
>
> Thanks for the constructive comments. We provide our responses as follows.
>
> ---
>
> **Q1: Fine-grained Prediction**
>
> We appreciate your insightful observations. Our method ensures fine-grained temporal prediction through the following design choices:
>
> - We utilize **2 FPS** for video processing, which offers relatively fine temporal resolution. This aligns with standard practices in video temporal grounding (VTG)—e.g., SnAG [1] (1 FPS) and UniVTG [2] (0.5 FPS).
> - For long videos, we adopt a **coarse-to-fine** grounding strategy in terms of temporal resolution to effectively distinguish closely adjacent events.
> - Our framework supports arbitrary temporal resolutions and can be adapted for finer precision. To validate this, we conducted experiments on Charades at 4 FPS, achieving slightly improved performance:
>
>     |FPS|R1\@0.5|R1\@0.7|mIoU|
>     |-|-|-|-|
>     |2 (default)|74.33|53.71|63.15|
>     |4|75.27|54.89|63.97|
>
> ---
>
> **Q2:  Query-guided Frame Importance Estimation**
>
> We appreciate your valuable suggestion. A lightweight module for dynamic frame importance estimation—*enabling adaptive token allocation*—is indeed a promising direction. We fully acknowledge the potential of this idea and plan to explore it in future research.
>
> ---
>
> **Q3: Timestamp Retrieval vs. Explicit Temporal Alignment**
>
> We appreciate your insightful suggestion regarding explicit temporal alignment supervision. After careful consideration, we opted not to include such supervision for the following reasons:
>
> - **Inherent Multimodal Retrieval Capabilities in MLLMs:**
>     - **Modern MLLMs exhibit strong multimodal retrieval abilities without explicit supervision**, as evidenced by tasks like *Needle-in-a-Haystack (NIAH)* [3,4], which require models to accurately retrieve target images ("needles") within long videos ("haystacks").
>     - By reformulating temporal grounding as a **timestamp retrieval** task using timestamp-interleaved sequences, we leverage and extend these inherent capabilities to temporal grounding, eliminating the need for auxiliary alignment supervision.
>
> - **Theoretical Equivalence to Explicit Alignment Methods:**
>     - Existing methods (e.g., GeLM [5]) employ alignment supervision by optimizing a similarity matrix between special tokens and visual features, ultimately enabling the token to effectively **retrieve query-relevant visual features**.
>     - Although our generative method differs from discriminative methods based on explicit alignment, it follows a **similar principle**: accurately retrieving timestamp tokens that correspond to query-relevant visual features, thereby achieving comparable grounding without requiring explicit contrastive supervision.
>
> ---
>
> **Q4: Unfreezing the Visual Backbone During Training**
>
> We appreciate your valuable suggestion. To investigate its impact, we conducted experiments where we unfroze the visual backbone of `Qwen2-VL-2B` (using LoRA) during training and evaluated performance on Charades-STA:
>
> |Vision Encoder|R1\@0.5|R1\@0.7|mIoU|
> |-|-|-|-|
> |Freeze|65.38|42.18|57.25|
> |Fine-tune|67.72|45.43|58.89|
>
> These findings **validate your insight** and suggest that our current approach could potentially benefit from this modification. Accordingly, we **plan to incorporate this adjustment** when retraining our UniTime-Full model.
>
> ---
>
> **Q5: Latency and Computational Cost**
>
> Thank you for your valuable feedback. While Table 4 in our original submission demonstrates the performance advantages of our multi-stage approach, we have now conducted a comprehensive computational analysis to quantify the associated overhead. Using the *calflops* library [6], we measured the latency, TFLOPs, and TMACs per query for videos from the Ego4D dataset, comparing multi-stage and single-stage inference:
>
> |Method|Latency|TFLOPs|TMACs|mIoU|
> |-|-|-|-|-|
> |Multi-stage|9.54s|326.88|164.94|17.25|
> |Single-stage|1.83s|91.19|45.59|9.83|
>
> - The multi-stage approach achieves **1.8× higher mIoU** (17.25 vs. 9.83)
> - This improvement comes with **5.2× higher latency** and **3.6× greater computational cost** (TFLOPs and TMACs)
>
> This analysis reveals the fundamental trade-off between grounding accuracy and computational efficiency in our method.  Future work could explore efficiency optimizations.
>
> ---
>
> **Q6: Impact of Frame Thresholds and Token Budgets**
>
> We appreciate your insightful questions regarding these key hyperparameters. To systematically evaluate their effects, we conducted comprehensive ablation studies:
>
> - **Threshold $N_f^{long}$**
>     - **Role**: Determines the maximum video length that can be processed directly
>         - If video length ≤ $N_f^{long}$: Process directly.
>         - If video length > $N_f^{long}$: Split into segments, process separately, and aggregate results.
>
>     - **Evaluation on Ego4D-NLQ**:
>
>         | $N_f^{long}$   | R1\@0.3 | R1\@0.5 | mIoU  |
>         | -------------- | ------- | ------- | ----- |
>         | 1024 (default) | 24.79   | 16.83   | 17.25 |
>         | 512            | 24.72   | 16.63   | 16.95 |
>         | 256            | 23.94   | 16.10   | 16.68 |
>
>     - **Key Findings**:
>         - The choice of $N_f^{long}$ has minimal impact on model accuracy because our method segments over-length videos and aggregates results.
>         - Adaptive Frame Scaling enforces a fixed token budget per segment. Consequently, a smaller $N_f^{long}$ increases inference time due to more video segments and more input tokens.
>         - **Default Choice**: 1024 for optimal efficiency-performance balance.
>
> - **Threshold $N_f^{short}$**
>     - **Role**: Decides single- or multi-stage processing:
>         - If video length ≤ $N_f^{short}$: Single-stage
>         - If video length > $N_f^{short}$: Multi-stage (coarse-to-fine)
>     - **Evaluation on TaCoS**:
>         | $N_f^{short}$ | mIoU (64-128s) | mIoU (128-256s) | Avg. mIoU |
>         | ------------- | -------------- | --------------- | --------- |
>         | 64            | 58.43 (multi)  | 54.57 (multi)   | 50.28     |
>         | 128 (default) | 56.59 (single) | 54.57 (multi)   | 50.00     |
>         | 256           | 56.59 (single) | 42.90 (single)  | 46.82     |
>
>     - **Key Findings**:
>         - For 64-128s videos: Minimal performance difference between multi-stage ($N_f^{short}$=64) and single-stage ($N_f^{short}$>=128) processing (ΔmIoU=1.84). However, the lower threshold increases average inference time as it forces more videos into multi-stage processing.
>         - For 128-256s videos: Multi-stage processing yields significantly better results (54.57 vs 42.90 mIoU)
>         - **Default Choice**: 128 to balance performance and efficiency.
>
> - **Token Budget**
>     - **Evaluation on Ego4D**:
>         | Token Budget    | R1\@0.3 | R1\@0.5 | mIoU  |
>         | --------------- | ------- | ------- | ----- |
>         | 16384 (default) | 24.79   | 16.83   | 17.25 |
>         | 8192            | 24.08   | 16.10   | 16.69 |
>         | 4096            | 23.04   | 15.07   | 15.96 |
>
>     - **Key Findings**:
>     Performance improves consistently with higher token budgets (up to GPU limits).
>
> ---
>
> **Q7: Multiple Valid Grounding Regions**
>
> For queries associated with multiple valid temporal moments (e.g., in QVHighlights), we incorporate all ground truth moments in chronological order within the training sequence. By doing so, the model learns to accurately handle ambiguous queries that may correspond to multiple grounding regions or moments. During inference, **UniTime can naturally predict multiple valid moments** for such queries, maintaining consistency with the training paradigm.
>
> ---
>
> We will incorporate the analysis, along with additional experiments, in our final draft.
>
> ---
>
> [1]: Fangzhou Mu, Sicheng Mo, and Yin Li. "Snag: Scalable and accurate video grounding." In CVPR, 2024.
>
> [2]: Lin, Kevin Qinghong, et al. "Univtg: Towards unified video-language temporal grounding." In ICCV, 2023.
>
> [3]: Zhang, Peiyuan, et al. "Long context transfer from language to vision." arXiv:2406.16852, 2024.
>
> [4]: Zhao, Zijia, et al. "Needle in a video haystack: A scalable synthetic framework for benchmarking video mllms." arXiv:2406.09367, 2024.
>
> [5]: Chen, Qirui, Shangzhe Di, and Weidi Xie. "Grounded multi-hop videoqa in long-form egocentric videos." In AAAI, 2025.
>
> [6]: Ye, Xiaoju. "calflops: a FLOPs and params calculate tool for neural networks in pytorch framework." 2023.

---

### Official Review · Reviewer_A5nc · 2025-06-29

**Clarity:** 3
**Significance:** 2
**Originality:** 3
**Rating:** 5
**Confidence:** 5

**Summary:**

This paper proposes UniTime, an MLLM-based model for universal video temporal grounding across diverse video scenarios. UniTime interleaves timestamp tokens with video tokens to enable precise boundary prediction and employs adaptive frame scaling to handle videos of different lengths, balancing spatial resolution and computational efficiency. For long videos, UniTime adopts a coarse-to-fine strategy—first retrieving relevant segments at low granularity, then refining boundaries within selected segments—achieving robust performance on both short and long videos. The proposed UniTime achieves superior performance on mainstream VTG and VideoQA benchmarks.

**Questions:**

Please refer to the above **[Weaknesses]** section. I suggest that the author provide more detailed comparisons and analysis in the experimental section. If the author can address my concerns, I would be happy to increase the score.

**Ethical Concerns:**

["NO or VERY MINOR ethics concerns only"]

**Final Justification:**

The rebuttal has addressed most of my concerns including:

* Regarding the preservation of general capabilities, the proposed method provides a plug-and-play module that enhances the VTG ability of MLLMs with only a small training cost. Since LoRA can be easily detached, the model's general capabilities are not harmed, which makes sense.

* For the ablations of key modules, the authors gradually added IG, AS, and SR modules based on Qwen2-VL-7B, showing consistent performance improvements with +16.79 mIoU over the baseline, which demonstrates the effectiveness of the method.

* Further enhancing the VTG capability of VideoQA models. Although it cannot be directly compared with VideoQA methods, the proposed approach can further improve the performance of VideoQA-type methods.

* Strong generalizability. The supplementary experiments provided by the authors verified significant improvements across different model sizes in the Qwen-VL series and Intern-VL series.

* Comparable to closed-source models. Compared to existing SOTA closed-source MLLMs, UniTime-Full achieved the best Avg. mIoU, approaching the corresponding SOTA closed-source models in both long and short video localization.

The author's rebuttal is very convincing, so I am willing to increase my initial score. This paper appears to be technically solid and will have a positive impact on the VTG field.

**Limitations:**

yes

**Paper Formatting Concerns:**

NIL

**Quality:**

3

**Strengths And Weaknesses:**

**[Strengths]**
1. The paper is written clearly, fluently, and is easy to follow

2. The proposed coarse-to-fine strategy, adaptive video frame sampling, and timestamp interleaving sequence methods are simple yet effective

3. The proposed MLLM-based UniTime method achieves impressive performance on multiple benchmarks


**[Weaknesses]**

**Important**
1. Lack of baseline comparison experiments. The proposed UniTime method is based on continued training of the Qwen2-VL-7B base model, but the paper does not provide the performance of Qwen2-VL-7B/Qwen2.5-VL-7B on VTG and VideoQA tasks. The authors should provide baseline experimental results and analyze how the main modules such as "Adaptive Scaling," "Multi-stage Inference," and "Segment Retrieval" improve upon the baseline

2. Lack of comparison with SOTA methods. Some recent major works, such as VideoChat-Flash [23], achieved a score of 64.7 on LongVideoBench and 74.7 on MLVU, which are better than the results presented in this paper.

3. No verification of whether general video understanding capabilities are maintained. UniTime utilizes the powerful video understanding capabilities of MLLM to help achieve better performance on VTG tasks, but does this training process harm the model's general video/visual understanding capabilities?

(Optional)

4. Flexibility verification. As a plug-and-play module, does it have generalizability across different baseline models (InternVL series) or models of different sizes (3B, 32B)? The authors should conduct more in-depth exploration of its generalizability

5. Closed-source model testing. The authors should test mainstream closed-source models (such as Claude, Gemini, Doubao, etc.) to evaluate whether the proposed UniTime method surpasses or narrows the gap with closed-source models on VTG tasks

---

> ### Author Rebuttal · Authors · 2025-07-31
>
> Thank you for your constructive comments. We have provided our responses below.
>
> ---
>
> **Q1: General Video Understanding Capabilities**
>
> - Our goal is to **enhance temporal grounding** by leveraging their strong multimodal understanding. While task-specific fine-tuning may reduce generality, our focus is on improving temporal reasoning, which is critical for many downstream applications.
>
> - UniTime uses **pluggable LoRA parameters**, allowing the base MLLM’s original video understanding capabilities to be fully restored by detaching LoRA. This is exactly how we conduct VideoQA evaluation in Table 6 in our original submission: first perform temporal grounding with UniTime-Full, then detach LoRA and answer questions using the grounded video segment.
>
> ---
>
> **Q2: Baseline Comparison Experiments**
>
> Thank you for your feedback. Below, we address the raised points in detail.
>
> - **Baseline Comparison on VTG Task**
>
>     |Method|Ego4D mIoU|TaCoS mIoU|Charades mIoU|ANet-Captions|
>     |-|-|-|-|-|
>     |Qwen2-VL-7B|0.46|2.64|8.98|5.97|
>     |Qwen2.5-VL-7B|0.82|5.29|52.42|21.85|
>     |UniTime-SP (Qwen2)|17.25|45.02|63.15|52.38|
>     |UniTime-SP (Qwen2.5)|18.80|50.00|64.55|51.41|
>     |UniTime-Full (Qwen2)|18.80|50.00|64.55|51.41|
>
>     Here, we evaluate the Video Temporal Grounding (VTG) performance of `Qwen2-VL-7B` and `Qwen2.5-VL-7B` across multiple benchmarks. Both variants of our UniTime model, UniTime-SP (dataset-specific setting) and UniTime-Full (universal setting), **demonstrate significantly superior performance compared to the Qwen-VL baselines**, especially on long-form video benchmarks.
>
> - **Baseline Comparison on VideoQA Task**: Comparisons on video question-answering (VideoQA) are presented in Table 6 in our original submission:
>     - The first row, "Uniform Sample", denotes the `Qwen2-VL-7B` baseline that uniformly samples 32 frames.
>     - The last row, "UniTime-Full", also stated in Q1, first grounds the relevant video segment with UniTime-Full, detaches LoRA, and then answers questions on the grounded clip.
>     - **UniTime significantly enhances the baseline MLLM's VideoQA accuracy** (e.g., +6.43 QA accuracy on CG-Bench).
>
> - **Ablation on Main Modules**:
>     In the main draft, **Table 4** presents our module ablation study with full implementation details in **Appendix A.2.1**. Here, we are happy to systematically analyze how each component enhances VGT performance upon the baseline:
>
>     |Configurations|R1\@0.3|R1\@0.5|mIoU|
>     |-|-|-|-|
>     |Qwen2-VL-7B|0.48|0.17|0.46|
>     |+Finetune|14.25|7.54|9.83|
>     |+Iterative Grounding|18.42|12.13|12.78|
>     |+Adaptive Scaling|17.91|11.99|12.39|
>     |+Segment Retrieval|24.79|16.83|17.25|
>
>     The above results were evaluated on Ego4D-NLQ:
>     - ``Qwen2-VL-7B``: The baseline sparsely samples frames from long videos, yielding poor accuracy (0.46 mIoU) caused by significant visual information loss.
>     - ``+Finetune``: While task-specific finetune delivers +9.37 mIoU gain, it retains sparse sampling limitations, remaining **suboptimal** (-7.42 mIoU compared to the last row).
>     - ``+Iterative Grounding``: Multi-stage grounding **improves accuracy** (+2.95 mIoU vs. finetuned baseline). However, it still suffers from the **omission of keyframes** due to temporally sparse sampling, resulting in the inability to generate accurate coarse-grained predictions (-4.70 mIoU compared to the last row).
>     - ``+Adaptive Scaling``: Dense sampling with token reduction slightly decreases performance (-0.39 mIoU) but is **essential for enabling Segment Retrieval**, wtih sufficient frames sampled.
>     - ``+Segment Retrieval``: Our full approach implements a *coarse-to-fine grounding* strategy that first identifies relevant video segments at a coarse level and then performs precise grounding within these segments. This delivers **substantial performance gains** (+16.79 mIoU over baseline, +4.86 mIoU over Row 4).
>
> ---
>
> **Q3: Comparison with SOTA VideoQA Methods**
>
> Thank you for your question. We’d like to clarify a few key points:
> - UniTime is specifically designed for video temporal grounding (VTG) and is **not directly comparable** to end-to-end VideoQA methods.
> - In Table 6 (in our original submission), we use VideoQA benchmarks solely to evaluate how effectively different VTG models identify question-relevant video segments.
> - UniTime can be **seamlessly integrated** with various Video-LLMs to enhance their performance by improving temporal grounding, reducing noise from irrelevant frames, and focusing computation on critical moments.
>
> Below, we evaluate using `VideoChat-Flash@224` as the base VideoQA model:
> - To ensure fair comparison under identical environments, we first reproduce the baseline results.
> - UniTime's temporal grounding provides consistent improvements over the reproduced baseline (+2.1 on LongVideoBench, +1.1 on MLVU).
>
> |Method|LongVideoBench|MLVU|
> |-|-|-|
> |Reported Results|64.2|74.5|
> |Reproduced Baseline|62.2|74.8|
> |+UniTime Grounding|64.3|75.9|
>
> ---
>
> **Q4: Flexibility Verification**
>
> Thank you for your interest in the generalizability of our approach. In Appendix G, we provided preliminary experiments with different model sizes of `Qwen2-VL` and `Qwen2.5-VL`. To further validate flexibility, we conducted additional experiments by integrating **UniTime** with various scales of `InternVL-2.5` models:
>
> |Base Model|Ego4D mIoU|Charades mIoU|
> |-|-|-|
> |Qwen2-VL-2B|0.42|6.70|
> |Qwen2-VL-2B + UniTime|7.29 `(+6.87)`|57.25 `(+50.55)`|
> |Qwen2-VL-7B|0.46|8.98|
> |Qwen2-VL-7B + UniTime|17.25 `(+16.79)`|63.15 `(+54.17)`|
> |Qwen2.5-VL-7B|0.82|52.42|
> |Qwen2.5-VL-7B + UniTime|16.61 `(+15.79)`|63.62 `(+11.20)`|
> |InternVL-2.5-2B|0.42|9.30|
> |InternVL-2.5-2B + UniTime|7.46 `(+7.04)`|59.03 `(+49.73)`|
> |InternVL-2.5-8B|1.77|17.54|
> |InternVL-2.5-8B + UniTime|12.03 `(+10.26)`|60.50 `(+42.96)`|
>
> The results demonstrate that our approach serves as an effective **plug-and-play module**, adapting well to different architectures and scales. For the final draft, we plan to extend testing to include a wider variety of models.
>
> ---
>
> **Q5: Closed-source Model Evaluation**
>
> Thank you for the valuable suggestion. To evaluate closed-source models, we conducted temporal grounding experiments on a randomly sampled subset of 100 videos from both the Ego4D and Charades datasets. It's important to note that we do not have insight into the training data used by these closed-source models, which could potentially result in data leakage.
>
> The results are summarized below:
>
> |Model|Ego4D mIoU|Charades mIoU| Avg. mIoU|
> |-|-|-|-|
> |Gemini-2.5-flash|13.9|38.7|26.3|
> |Gemini-2.5-pro|**20.5**|39.3|29.9|
> |GPT-4.1-mini|5.0|31.5|18.3|
> |GPT-4o|8.0|43.8|25.9|
> |Doubao-1-5-thinking-vision-pro-250428|13.5|**72.2**|42.9|
> |UniTime-Full|17.2|69.5|**43.4**|
>
> - `Doubao` achieved the best performance on short videos (72.2 Charades mIoU).
> - `Gemini-2.5-pro` showed superior grounding capability for long videos (20.5 Ego4D mIoU).
> - These closed-source models struggle to achieve high performance concurrently on both long and short videos, indicating a lack of generalization.
> - UniTime achieves competitive performance compared to these closed-source models while maintaining more balanced capabilities for both short and long-form video grounding benchmarks, proving our approach to be overall the best in terms of consistent performance across varying video types.
>
> ---
>
> We sincerely appreciate these valuable comments. Based on your suggestion, we will incorporate these comprehensive evaluation results and analyses in our final manuscript revision.

---

> > ### Comment · Reviewer_A5nc · 2025-08-05
> >
> > Thanks to the authors for the detailed reply and extensive experiments. After carefully reading the rebuttal, most of my concerns have been addressed:
> >
> > * Regarding the preservation of general capabilities, the proposed method provides a plug-and-play module that enhances the VTG ability of MLLMs with only a small training cost. Since LoRA can be easily detached, the model's general capabilities are not harmed, which makes sense.
> >
> > * For the ablations of key modules, the authors gradually added IG, AS, and SR modules based on Qwen2-VL-7B, showing consistent performance improvements with +16.79 mIoU over the baseline, which demonstrates the effectiveness of the method.
> >
> > * Further enhancing the VTG capability of VideoQA models. Although it cannot be directly compared with VideoQA methods, the proposed approach can further improve the performance of VideoQA-type methods.
> >
> > * Strong generalizability. The supplementary experiments provided by the authors verified significant improvements across different model sizes in the Qwen-VL series and Intern-VL series.
> >
> > * Comparable to closed-source models. Compared to existing SOTA closed-source MLLMs, UniTime-Full achieved the best Avg. mIoU, approaching the corresponding SOTA closed-source models in both long and short video localization.
> >
> > Given that the authors have addressed my concerns raised before the rebuttal (baseline comparisons, SOTA comparative experiments, and preservation of general capabilities, etc), I am willing to improve my initial score. Please include the supplementary experiments and analyses from the rebuttal phase in the final version.

---

> > > ### Author Response · Authors · 2025-08-06
> > >
> > > We greatly appreciate the time you took to review our work and for raising your rating. We will carefully reflect on your suggestions and incorporate them into the final version.

---

### Official Review · Reviewer_Zb87 · 2025-07-03

**Clarity:** 4
**Significance:** 4
**Originality:** 4
**Rating:** 5
**Confidence:** 4

**Summary:**

The paper proposes an approach for temporal grounding of videos where the short and long videos are treated in a unified framework. In particular, keeping the token budget fixed, spatial granularity is scaled accordingly (lower for long videos; and vice versa). The paper, in addition, leverages techniques such as interleaved timestamp tokens, coarse-to-fine search scheme, and train-time optimization to push the performance over existing MLLM-based approaches

**Questions:**

see weaknesses

**Ethical Concerns:**

["NO or VERY MINOR ethics concerns only"]

**Final Justification:**

The authors addressed my concerns on (a) reporting the pre-training datasets of prior approaches to draw head-to-head comparisons, and (b) the ablation on explicit timestamp insertion (proposed) vs. positional encodings.

I suggest the authors include the above in the final version of the paper.

**Limitations:**

yes

**Paper Formatting Concerns:**

appendices are included in the main submission

**Quality:**

4

**Strengths And Weaknesses:**

### Strengths
- Writing and presentation of the results in the paper is clear
- Ablation study is highly informative of the design choices


### Weaknesses
- Highlighting the pre-training dataset (type, quantity, etc.) of the related works in Table 3 would have been helpful to draw more nuanced conclusions
- An ablation on the design choice of lines 120-121 would be great: “we leverage the retrieval capabilities of MLLMs to read out the inserted timestamp tokens, rather than decoding dense position encodings”. Or point to an existing work that did
- Eq 2 doesn’t contain “end” timestamps of the segments. Do the authors have an impression of whether such an input would be effective?

Minor
- Negative numbers are highlighted in green, giving a false first impression that they are positive

---

> ### Author Rebuttal · Authors · 2025-07-31
>
> Thanks for your positive comments! We provide the feedback as follows.
>
> ---
>
> **Q1: Pre-training Datasets of the Related Works**
>
> Thank you for your valuable suggestion. Here, we summarize the pre-training datasets used by the related works. Some of the methods (like TimeMarker) are trained on a dozen datasets. Due to space constraints, we will provide the complete list in the final draft.
>
> |Method|Pre-training datasets|Sample|Duration|
> |-|-|-|-|
> |UniVTG|Ego4D, VideoCC, YouTubeHL, ...|4.3M|Long & Short
> |VTG-LLM|VTG-IT-120K, Charades, Youcook2, ...|97K|Short|
> |Momentor|Moment-10M|10M|Long|
> |VTimeLLM|InternVid-10M-FLT, VideoInstruct100K, DiDeMo, ...|170K|Short|
> |Timechat|TimeIT (DiDeMo, QueryD, HiREST, ...)|125K|Short|
> |TimeMarker|ShareGPTVideo, ShareGPT4Video, VideoChat2-IT, ...|5.4M|Long & Short|
> |TimeSuite|TimePro (DiDeMo, QueryD, HiREST, ...)|349K|Long & Short|
> |UniTime|NaQ, Momentor, Ego4d-NLQ, ...|1.3M|Long & Short|
>
> ---
>
> **Q2: Timestamp Token Insertion vs. Dense Position Encodings**
>
> Thank you for your insightful question. To address this, we conducted ablation experiments on Charades:
>
> |Method|R1\@0.5|R1\@0.7|mIoU|
> |-|-|-|-|
> |Dense Position Encodings|48.44|27.15|44.85|
> |Timestamp Token Insertion (UniTime-SP)|74.33|53.71|63.15|
>
> - Our base model, `Qwen2-VL-7B`, utilizes Multimodal Rotary Position Embedding (MRoPE), which decomposes position embeddings into temporal, height, and width components. After fine-tuning, temporal grounding remains **suboptimal with dense position encodings**.
> - In contrast, **timestamp token insertion yields significant gains**. We attribute this improvement to the MLLM's strong multimodal retrieval ability and the simplification of the task into (1) identifying question-relevant video segments and (2) directly reading their corresponding timestamp tokens.
>
> ---
>
> **Q3: The “end” Timestamps of the Segments**
>
> In the segment retrieval (coarse-grained grounding) stage, the model is provided with the starting timestamps of each segment, denoted as *{$t_1$, $t_2$, ..., $t_N$}*. The model predicts a segment boundary *$t_i$*, and the corresponding segment interval is defined as *[$t_i$, $t_{i+1}$]*. For the final segment, the end timestamp *t$_{i+1}$* is set to the total duration of the video.
>
> ---
>
> **Q4: The Color of Negative Numbers**
>
> Thank you for bringing this to our attention. We acknowledge the need for improved visual distinction and will use a distinct color for negative numbers in the revised manuscript.
>
> ---
>
> **Q5: Inclusion of Appendices in the Main Submission**
>
> We appreciate your question regarding appendix placement. In accordance with this year's NeurIPS submission guidelines, which state that *“authors may optionally choose to include some or all of the technical appendices in the same PDF above.”*, we have incorporated the most substantive appendices into our main submission.
>
> ---
>
> We will incorporate the above clarifications, experiments, and analyses into our final draft.

---

> > ### Comment · Reviewer_Zb87 · 2025-08-07
> > **Thanks**
> >
> > Thanks you for the rebuttal.
> >
> > - In regards to Q1, do the authors have any intuition on how the performance would scale w.r.t. (a) the number of data samples used in training, and (b) the inclusion of different lengths of videos. Or more broadly, given the current state of results, how would the authors approach the task in order to further improve performance?

---

> > > ### Author Response · Authors · 2025-08-08
> > >
> > > Thank you for your questions!
> > >
> > > - **Data Scaling**.
> > > UniTime-Full's performance improves consistently with increased training data, as shown below:
> > >     | Data Percentage | Ego4D mIoU | Charades mIoU |
> > >     | --------------- | ---------- | ------------- |
> > >     | 20\%            | 14.5       | 52.0          |
> > >     | 40\%            | 17.0       | 56.2          |
> > >     | 60\%            | 18.3       | 61.6          |
> > >     | 80\%            | 18.3       | 64.3          |
> > >     | 100\%           | 18.8       | 64.6          |
> > >
> > > - **Video Length**.
> > > While we have not yet tested the impact of varying video lengths due to time constraints, we hypothesize that incorporating diverse durations during training could enhance model generalization by offering varied temporal dynamics.
> > >
> > >
> > > - **Future Work**.
> > > To further improve performance, we consider the following directions:
> > >     1. **Balanced Sampling** – Ensuring a well-distributed dataset by balancing topics, durations, and view counts.
> > >     2. **Extended Capabilities** – Expanding training datasets to support additional capabilities such as Highlight Detection, Video Summarization, and Video Dense Captioning.
> > >     3. **Query-guided Frame Importance Estimation** – As suggested by Reviewer F2GW, optimizing token allocation through frame importance estimation is a promising avenue.
> > >     4. **Full Fine-tuning** – In response to Reviewer F2GW’s Q4, unfreezing the visual backbone could further boost performance.
> > >
> > > Please don't hesitate to reach out if any further questions or suggestions arise.

---

### Official Review · Reviewer_YJyw · 2025-07-03

**Clarity:** 2
**Significance:** 3
**Originality:** 2
**Rating:** 4
**Confidence:** 3

**Summary:**

This paper introduces UniTime, a versatile Multimodal Large Language Model (MLLM) designed for universal video temporal grounding. The primary goal of UniTime is to accurately localize events within videos of varying lengths, genres, and viewpoints based on natural language queries. There are three core contributions: 1) an adaptive frame scaling mechanism that dynamically adjusts frame resolution based on video duration to manage computational load while preserving semantic information; 2) a timestamp-interleaved sequence representation that explicitly integrates temporal information by interleaving textual timestamp tokens with visual frame tokens; and 3) a coarse-to-fine grounding strategy for long videos, which first identifies a broad temporal region of interest and then performs precise localization within that segment.

**Questions:**

Please list up and carefully describe questions and suggestions for the authors, which should focus on key points (ideally around 3–5) that are actionable with clear guidance. Think of the things where a response from the author can change your opinion, clarify a confusion or address a limitation. You are strongly encouraged to state the clear criteria under which your evaluation score could increase or decrease. This can be very important for a productive rebuttal and discussion phase with the authors.
1.	Clarifying the Novelty of the Timestamp-Interleaved Sequences. Could you please elaborate on the specific novelty of your timestamp-interleaved sequence representation compared to prior methods?
2.	Resolving the Contradiction in Frame/Token Compression. The manuscript states that token compression is preferable to frame resizing for preserving semantic information, which raises the question of why frame resizing is still a necessary component of your pipeline.
3.	Justification and Sensitivity Analysis for Adaptive Scaling Hyperparameters.

**Ethical Concerns:**

["NO or VERY MINOR ethics concerns only"]

**Final Justification:**

The author resolved some of my concerns but the paper still contains many weakness. I keep the score based on the authors response.

**Limitations:**

Yes

**Quality:**

3

**Strengths And Weaknesses:**

Please provide a thorough assessment of the strengths and weaknesses of the paper. A good mental framing for strengths and weaknesses is to think of reasons you might accept or reject the paper. Please touch on the following dimensions: Quality, Clarity, Significance, and Originality. For more information, please see the NeurIPS 2025 Reviewer Guidelines (https://neurips.cc/Conferences/2025/ReviewerGuidelines). You can incorporate Markdown and LaTeX into your review. See https://openreview.net/faq.

1.	Strengths
a.	Well-Designed and Practical Method: The proposed UniTime framework effectively combines several well-motivated ideas. The adaptive frame scaling is a practical solution to the context-length limitations of MLLMs. The coarse-to-fine inference strategy is an intuitive and effective way to handle long videos hierarchically.
b.	Comprehensive Evaluation and Exceptional Results: The experimental setup is a major strength of this work including diverse benchmarks, multiple settings and robustness tests. The paper presents very strong and consistent results across a wide array of benchmarks. UniTime substantially outperforms previous state-of-the-art models in all three evaluation settings: dataset-specific fine-tuning, universal, and zero-shot.
2.	Weaknesses
a.	Timestamp-Interleaved Novelty. The paper acknowledges that other recent models like Mr.BLIP [31], TimeMarker [6], and VideoLLaMA3 [56] also use explicit textual timestamps. The manuscript should highlight specific novelty of UniTime's timestamp-interleaved sequence representation in comparison to these prior works.
b.	Ambiguity in the Rationale for Frame Resizing versus Token Compression: There appears to be a point of confusion in the justification for using both frame resizing and token compression. The paper states (L102-104): “Unlike frame resizing, which may degrade spatial details, token compression reduces redundancy at the token level while better preserving key semantic information…”. This statement raises the question of why frame resizing is necessary if token compression is a superior method for preserving semantic information.
c.	Lack of Justification for Hyperparameter Choices: The adaptive frame scaling mechanism is governed by two critical thresholds, N_f^short  and N_f^long , which are set to 128 and 1024, respectively. The paper does not provide a justification or an ablation study for these specific hyperparameter choices. The model's performance may be sensitive to these values, and their selection appears somewhat arbitrary.

---

> ### Author Rebuttal · Authors · 2025-07-28
>
> Thank you for your constructive comments. Our responses are provided below.
>
> ---
> **Q1: Novelty of Timestamp-Interleaved Sequences**
>
> Thank you for the feedback. We would like to clarify the key novelty of UniTime’s timestamp-interleaved sequence representation as follows:
>
> - **Dynamic Timestamp Resolution:** Unlike prior methods that rely on fixed per-frame insertion, UniTime introduces dynamic resolution control. Timestamp tokens can be interleaved at multiple scales—either per frame for fine-grained predictions or per segment for coarse-grained reasoning. This adaptability enhances flexibility in temporal modeling (see Section 2.2, Timestamp-Interleaved Sequence Construction).
>
> - **Multi-Scale Temporal Predictions:** Coupled with adaptive frame scaling, UniTime supports coarse-to-fine inference over varying video lengths. This improves grounding accuracy on long videos, addressing limitations of prior methods that rely on sparse sampling or short clips.
>
>
> ---
>
> **Q2: Explanation for Frame Resizing and Token Compression Choices**
>
> Thank you for the insightful comments. To resolve the ambiguity, we conducted an ablation study on the Ego4D-NLQ dataset, evaluating the effects of different video processing strategies on grounding performance and computational efficiency:
>
> |Short Video|Long Video|R1\@0.3|R1\@0.5|mIoU|Seg Retrieval|Training Time|
> |-|-|-|-|-|-|-|
> |frame resize (default)|token compression|24.79|16.83|17.25|45.28|21 h|
> |frame resize|frame resize|18.53|12.67|12.88|35.36|12 h|
> |token compression|token compression|24.06|16.18|16.71|45.28|24.5 h|
>
> Our findings indicate that:
>
> - **Long Video:** Adaptive Frame Scaling reduces tokens per frame while keeping the frame rate fixed. Though some visual cues are lost, this feature-level token compression retains more semantic information than frame resizing, which discards input resolution. This leads to better segment retrieval accuracy and recall (Rows 1 vs. 2).
>
> - **Short Video:** For short clips with sufficient tokens, both methods perform similarly (Rows 1 vs. 3). However, token compression requires high-resolution inputs before merging, incurring higher computational costs. In UniTime-Full, using token compression for short videos increased training time by 5 days compared to frame resizing.
>
> Therefore, instead of a unified strategy, we adopt a hybrid approach to balance performance and efficiency: frame resizing for short videos (for computational efficiency) and token compression for long videos (to preserve visual details). This design achieves strong performance with practical training costs.
>
> ---
>
> **Q3: Ablation Study of Adaptive Frame Scaling Hyperparameters**
>
> Thank you for the thoughtful feedback. We have conducted a systematic ablation study on two key hyperparameters:
>
> - **Threshold $N_f^{long}$**
>     - **Role**: Determines the maximum video length that can be processed directly
>         - If video length ≤ $N_f^{long}$: Process directly.
>         - If video length > $N_f^{long}$: Split into segments, process separately, and aggregate results.
>
>     - **Evaluation on Ego4D-NLQ**:
>
>         | $N_f^{long}$   | R1\@0.3 | R1\@0.5 | mIoU  |
>         | -------------- | ------- | ------- | ----- |
>         | 1024 (default) | 24.79   | 16.83   | 17.25 |
>         | 512            | 24.72   | 16.63   | 16.95 |
>         | 256            | 23.94   | 16.10   | 16.68 |
>
>     - **Key Findings**:
>         - The choice of $N_f^{long}$ has minimal impact on model accuracy because our method segments over-length videos and aggregates results.
>         - Adaptive Frame Scaling enforces a fixed token budget per segment. Consequently, a smaller $N_f^{long}$ increases inference time due to more video segments and more input tokens.
>         - **Default Choice**: 1024 for optimal efficiency-performance balance.
>
> - **Threshold $N_f^{short}$**
>     - **Role**: Decides single- or multi-stage processing:
>         - If video length ≤ $N_f^{short}$: Single-stage
>         - If video length > $N_f^{short}$: Multi-stage (coarse-to-fine)
>     - **Evaluation on TaCoS**:
>         | $N_f^{short}$ | mIoU (64-128s) | mIoU (128-256s) | Avg. mIoU |
>         | ------------- | -------------- | --------------- | --------- |
>         | 64            | 58.43 (multi)  | 54.57 (multi)   | 50.28     |
>         | 128 (default) | 56.59 (single) | 54.57 (multi)   | 50.00     |
>         | 256           | 56.59 (single) | 42.90 (single)  | 46.82     |
>
>     - **Key Findings**:
>         - For 64-128s videos: Minimal performance difference between multi-stage ($N_f^{short}$=64) and single-stage ($N_f^{short}$>=128) processing (ΔmIoU=1.84). However, the lower threshold increases average inference time as it forces more videos into multi-stage processing.
>         - For 128-256s videos: Multi-stage processing yields significantly better results (54.57 vs 42.90 mIoU)
>         - **Default Choice**: 128 to balance performance and efficiency.
>
> ---
>
> We will incorporate the above clarifications, experiments, and analyses into our final draft.

---

### Comment · Area_Chair_E9qj · 2025-08-05

Dear Reviewers,

Thank you very much again for performing this extremely valuable service to the NeurIPS authors and organizers.

As the authors have provided detailed responses, it would be greatly appreciated if you could take a moment to review them and see if your concerns have been addressed. Given that the discussion phase is nearing its end, your prompt feedback would be especially valuable, allowing the authors a final opportunity to offer any additional clarifications if needed.

Cheers,

AC

---

### Decision · Program_Chairs · 2025-09-17

**Decision:**

Accept (poster)

**Comment:**

This paper introduces a multimodal large language model for universal temporal grounding, which is a practical and important problem. There are various strengths of this work:

- The techniques are well-motivated and clearly designed (Reviewer YJyw)

- Evaluation is encyclopedic and robustly validates the strengths of the proposed method (Reviewer Zb87).

In addition to these strengths, there are certain concerns regarding general video understanding capabilities of the model (Reviewer A5nc) and technical flaws (Reviewer F2GW). However, during rebuttal, authors have satisfactorily resolved these concerns. In summary, I recommend an acceptance for this work.